# A New Perspective on "How Graph Neural Networks Go Beyond Weisfeiler-Lehman?"

**Asiri Wijesinghe & Qing Wang**
School of Computing, Australian National University, Canberra, Australia
`{asiri.wijesinghe, qing.wang}@anu.edu.au`

## Abstract

We propose a new perspective on designing powerful Graph Neural Networks (GNNs). In a nutshell, this enables a general solution to inject structural properties of graphs into a message-passing aggregation scheme of GNNs. As a theoretical basis, we develop a new hierarchy of local isomorphism on neighborhood subgraphs. Then, we theoretically characterize how message-passing GNNs can be designed to be more expressive than the Weisfeiler Lehman test. To elaborate this characterization, we propose a novel neural model, called *GraphSNN*, and prove that this model is strictly more expressive than the Weisfeiler Lehman test in distinguishing graph structures. We empirically verify the strength of our model on different graph learning tasks. It is shown that our model consistently improves the state-of-the-art methods on the benchmark tasks without sacrificing computational simplicity and efficiency.

## 1 Introduction

Many Graph Neural Networks (GNNs) employ a message-passing aggregation scheme to learn low-dimensional vector space representations for nodes in a graph (Kipf & Welling, 2017; Veličković et al., 2017; Hamilton et al., 2017; Gilmer et al., 2017; Sato, 2020; Loukas, 2020; de Haan et al., 2020). Let $G = (V, E)$ be a graph. For each node $v \in V$, a message-passing aggregation scheme recursively aggregates the feature vectors of nodes in the neighborhood of $v$ and combines the aggregated information with the feature vector of $v$ itself to obtain a representation. Since there is no natural ordering on nodes, such message-passing aggregation schemes are usually required to be permutation-invariant (Maron et al., 2018; Keriven & Peyré, 2019; Garg et al., 2020).

Despite advances of GNNs in various graph learning tasks such as node classification (Kipf & Welling, 2017; Xu et al., 2018), graph classification (Xu et al., 2019; Wu et al., 2019) and link prediction (Zhang & Chen, 2017), there is still a lack of theoretical understanding of how to design powerful and practically useful GNNs that can capture rich structural information of graphs. Recent studies (Xu et al., 2019; Morris et al., 2019) have explored the connections between GNNs and the Weisfeiler-Lehman (WL) test (Weisfeiler & Leman, 1968). By representing a neighborhood as a multiset of feature vectors and treating the neighborhood aggregation as an aggregation function over multisets, Xu et al. (2019) showed that message-passing GNNs are at most as powerful as the WL test in distinguishing graph structures. However, many simple graph structures still cannot be distinguished by the WL test, e.g., $G_1$ and $G_2$ shown in Figure 1. A question is: *how to design expressive yet simple GNNs that can go beyond the WL test with a theoretically provable guarantee?*

Recently, there have been three main directions of extending GNNs beyond WL: (1) building GNNs for higher-order WL (i.e. $k$-WL with $k \geq 3$) or variants (Maron et al., 2019; Morris et al., 2020; 2019); (2) counting on pre-defined substructures as additional features (Bouritsas et al., 2020); (3) augmenting node identifiers or random features into GNNs (You et al., 2021; Vignac et al., 2020; Sato et al., 2021). Unlike these works, we aim to introduce a general solution upon which GNNs can be enhanced to capture structural properties of graphs. This solution enables GNNs to *provably be more expressive* than the Weisfeiler-Lehman test, but still *computationally efficient*. It overcomes the following limitations of existing works. Compared with higher-order WL methods in (1) which require high computational overhead and are impractical, our method goes beyond the WL test but is still computationally efficient. Compared with the methods on counting substructures in (2), our

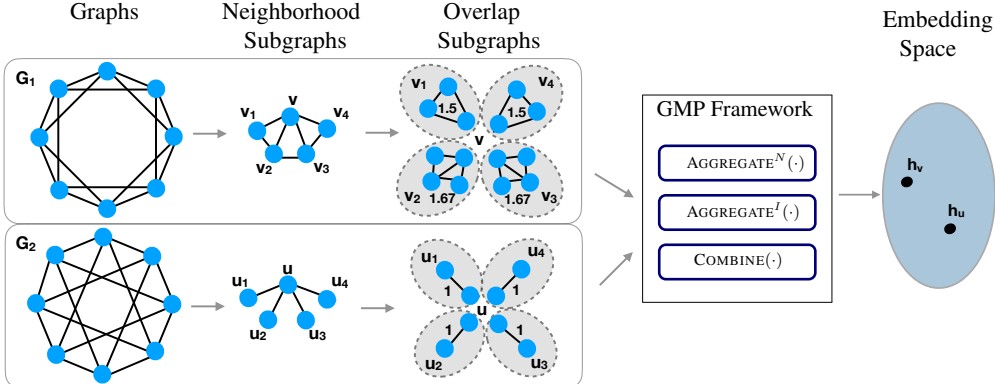

Figure 1: An overview of our proposed framework for GNNs that can go beyond the WL test in distinguishing non-isomorphic graphs $G_1$ and $G_2$. The overlap subgraphs of $G_1$ and $G_2$ are structurally different, which are captured by structural coefficients defined in Eq. 4.

method does not require to handcraft substructures. Compared with the methods of augmenting node identifiers or random features in (3), our method can flexibly quantify local structures (see examples in Figure 3) and also capture different classes of local structures w.r.t. different graph learning tasks.

Our work is grounded in three observations: (i) Treating a neighborhood as a multiset of feature vectors ignores the rich structure information among vertices in the neighborhood, thereby limiting the representational capacity of the model. Thus, we represent a neighborhood as a neighborhood subgraph in which vertices are structurally related, and show that the WL test is only as powerful as distinguishing neighborhood subgraphs in terms of their subtree structures in the neighborhood. (ii) There exists a natural class of isomorphic graphs, which strictly lies in between neighborhood *subgraph isomorphism* and neighborhood *subtree isomorphism*. We call it *overlap (subgraph) isomorphism*. The notion of overlap subgraph enables us to characterize structural interactions of vertices and inject them into a message-passing aggregation scheme for GNNs. (iii) By designing a proper function for quantifying structural interactions of vertices and preserving the injectiveness of a message-passing aggregation scheme, more expressive GNNs can be developed. We propose a new GNN model that is strictly more expressive than the WL test to demonstrate an instance of this kind.

**Contributions.** In summary, the main contributions of this work are as follows:

- We introduce a new hierarchy of local isomorphism to characterise different classes of local structures in neighborhood subgraphs, and discuss its connections with the WL test and GNNs (Section 2 and Theorems 1-2).

- We develop a simple yet powerful framework to inject structural properties into a message-passing aggregation scheme, and theoretically characterize how GNNs can be designed to be more expressive beyond the WL test (Section 3 and Theorem 3).

- We propose a novel neural model for graph learning, called GraphSNN, and prove that GraphSNN is strictly more expressive than the the WL test in distinguishing graph structures (Section 4 and Theorem 4).

- We show that, due to the way of injecting structural properties into a structured-message-passing aggregation scheme, GraphSNN can overcome the oversmoothing issue (Chen et al., 2020a; Zhao & Akoglu, 2019; Li et al., 2018) (Section 5.4).

We have conducted experiments on benchmark tasks (Hu et al., 2020). The experimental results show that our model is highly efficient and can significantly improve the state-of-the-art methods without sacrificing computational simplicity.

**Related work.** Weisfeiler-Lehman (WL) hierarchy is a well-established framework for graph isomorphism tests (Grohe, 2017). Introduced by Weisfeiler and Lehman (Weisfeiler & Leman, 1968), the Weisfeiler-Lehman algorithm (also called 1-WL or color refinement) is a computationally efficient heuristic for testing graph isomorphism (Babai & Kucera, 1979). It is known that k-WL is strictly more powerful than (k-1)-WL when k≥3 (Cai et al., 1992; Grohe, 2017).

Message-passing GNNs are typically considered as a differentiable neural generalization of the Weisfeiler-Lehman algorithms on graphs. It has been reported (Xu et al., 2019) that some popular GNNs such as GCN (Kipf & Welling, 2017) and GraphSAGE (Hamilton et al., 2017) are at most powerful as 1-WL in distinguishing graph structures. Xu et al. (2019) has shown that Graph Isomorphism Network (GIN) can be as powerful as 1-WL. At its core, GIN provides an injective aggregation scheme that is defined as a function over multisets of feature vectors, and thus GIN has the representational power to map any two different multisets of feature vectors to different representations in an embedding space.

A considerable amount of efforts has been devoted to improve the expressive power of GNNs beyond 1-WL. Generally, there are three directions: (1) Several works proposed higher-order variants of GNNs that are as powerful as k-WL with $k \geq 3$ (Azizian & Lelarge, 2020). For example, Morris et al. (2019) introduced k-order graph networks that are expressive as a set-based variant of k-WL, Maron et al. (2019) proposed a reduced 2-order graph network that is as expressive as 3-WL, and Morris et al. (2020) proposed a local version of k-WL which considers only a subset of vertices in a neighborhood. However, these more expressive GNNs are impractical to use due to their inherent high computational costs and sophisticated design. (2) Some works attempted to incorporate inductive biases based on isomorphism counting on pre-defined topological features such as triangles, cliques, and rings (Bouritsas et al., 2020; Liu et al., 2020; Monti et al., 2018), similar to the traditional ideas of graph kernels (Yanardag & Vishwanathan, 2015). However, pre-defining topological features requires domain-specific expertise, which is often not readily available. (3) Most recently, several works explored the ideas of augmenting GNNs using node identifiers or random features. For example, Vignac et al. (2020) proposed a method that maintains a "local context" for each node based on manipulating node identifiers in a permutation equivariant way. You et al. (2021) developed ID-GNNs by taking into account the identity information of vertices. Chen et al. (2020b) and Murphy et al. (2019) assigned one-hot IDs to vertices based on the ideas of relational pooling. Sato et al. (2021) added a random feature to each node to improve the representational capability of GNNs.

Our work is fundamentally different from existing models by injecting properties of structural interactions among vertices based on a natural class of isomorphic graphs in the local neighborhood (i.e., overlap subgraph isomorphism) into a message-passing aggregation scheme of GNNs.

## 2 A New Hierarchy of Local Isomorphism

In this section, we characterize a hierarchy of graph isomorphism based on local neighborhood subgraphs and explore its connections to 1-WL.

Let $G = (V, E)$ be a simple, undirected graph with a set $V$ of vertices and a set $E$ of edges. The set of neighbors of a vertex $v$ is denoted by $\mathcal{N}(v) = \{u \in V | (v, u) \in E\}$. The *neighborhood subgraph* of a vertex $v$, denoted by $S_v$, is the subgraph induced in $G$ by $\tilde{\mathcal{N}}(v) = \mathcal{N}(v) \cup \{v\}$, which contains all edges in $E$ that have both endpoints in $\tilde{\mathcal{N}}(v)$. For two adjacent vertices $v$ and $u$, i.e., $(v, u) \in E$, the *overlap subgraph* $S_{vu}$ between $v$ and $u$ is defined as $S_{vu} = S_v \cap S_u$.

Let $S_i$ and $S_j$ be the neighborhood subgraphs of two vertices $i$ and $j$ that are not necessarily adjacent, and $h_v$ be the feature vector of a vertex $v \in V$. In the following, we define three notions of isomorphism, which correspond to different classes of local structures in neighborhood subgraphs.

**Definition 1.** $S_i$ and $S_j$ are subgraph-isomorphic, *denoted as* $S_i \simeq_{subgraph} S_j$, *if there exists a bijective mapping* $g : \tilde{\mathcal{N}}(i) \to \tilde{\mathcal{N}}(j)$ *such that* $g(i) = j$ *and for any two vertices* $v_1, v_2 \in \tilde{\mathcal{N}}(i)$, $v_1$ *and* $v_2$ *are adjacent in* $S_i$ *iff* $g(v_1)$ *and* $g(v_2)$ *are adjacent in* $S_j$, *and* $h_{v_1} = h_{g(v_1)}$ *and* $h_{v_2} = h_{g(v_2)}$.

**Definition 2.** $S_i$ and $S_j$ are overlap-isomorphic, *denoted as* $S_i \simeq_{overlap} S_j$, *if there exists a bijective mapping* $g : \tilde{\mathcal{N}}(i) \to \tilde{\mathcal{N}}(j)$ *such that* $g(i) = j$ *and for any* $v' \in \mathcal{N}(i)$ *and* $g(v') = u'$, $S_{iv'}$ *and* $S_{ju'}$ *are subgraph-isomorphic.*

**Definition 3.** $S_i$ and $S_j$ are subtree-isomorphic, *denoted as* $S_i \simeq_{subtree} S_j$, *if there exists a bijective mapping* $g : \tilde{\mathcal{N}}(i) \to \tilde{\mathcal{N}}(j)$ *such that* $g(i) = j$ *and for any* $v' \in \tilde{\mathcal{N}}(i)$ *and* $g(v') = u'$, $h_{v'} = h_{u'}$.

Theorem 1 states that there is a hierarchy among these notions of local isomorphism on neighborhood subgraphs, where subgraph-isomorphism is the strongest one, subtree-isomorphism is the weakest,

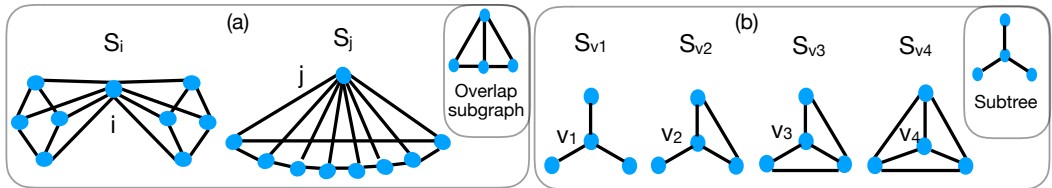

Figure 2: (a) $S_i$ and $S_j$ are overlap-isomorphic (i.e., having the same overlap subgraph) but not subgraph-isomorphic; (b) Four neighborhood subgraphs $\{S_{v_i}|i = 1, 2, 3, 4\}$ are subtree-isomorphic (i.e., having the same subtree) but not overlap-isomorphic.

and overlap-isomorphism lies in between. Figure 2 shows two groups of graphs: one is distinguishable w.r.t. subgraph-isomorphism but not overlap-isomorphism, while the other is distinguishable by overlap-isomorphism but not subtree-isomorphism.

**Theorem 1.** *The following statements are true: (a) If $S_i \simeq_{subgraph} S_j$, then $S_i \simeq_{overlap} S_j$; but not vice versa; (b) If $S_i \simeq_{overlap} S_j$, then $S_i \simeq_{subtree} S_j$; but not vice versa.*

Let $\mathcal{S} = \{S_v | v \in V\}$ and $\zeta : \mathcal{S} \to \mathbb{R}^d$ mapping each neighborhood subgraph in $\mathcal{S}$ into a node embedding in $\mathbb{R}^d$. The following theorem states that GNNs that are as powerful as 1-WL can distinguish two neighborhood subgraphs only w.r.t. subtree-isomorphism at each layer.

**Theorem 2.** *Let $M$ be a GNN. $M$ is as powerful as 1-WL in distinguishing non-isomorphic graphs if $M$ has a sufficient number of layers and each layer can map any $S_i$ and $S_j$ in $\mathcal{S}$ into two different embeddings (i.e., $\zeta(S_i) \neq \zeta(S_j)$) if and only if $S_i \not\simeq_{subtree} S_j$.*

The complete proofs of these theorems are provided in Appendix C.

## 3    A GENERALISED MESSAGE-PASSING FRAMEWORK

In this section, we present a generalised message-passing framework (GMP) which enables to inject local structure into an aggregation scheme, in light of overlap subgraphs. We theoretically characterize how GNNs can be designed to be more expressive than 1-WL in this framework.

Let $\mathcal{S}^* = \{S_{vu} | (v, u) \in E\}$ be the set of overlap subgraphs in $G$. We define *structural coefficients* for each vertex $v$ and its neighbors, i.e., $\omega : \mathcal{S} \times \mathcal{S}^* \to \mathbb{R}$ such that $A_{vu} = \omega(S_v, S_{vu})$. A question arising is: what are the desirable properties of such a function $\omega$? Ideally, it should quantify how a vertex $v$ structurally interacts with its neighbor $u$ in the local neighborhood. Thus, given $S_{vu} = (V_{vu}, E_{vu})$ and $S_{vu'} = (V_{vu'}, E_{vu'})$, a carefully designed $\omega$ should exhibit the following properties:

(1) **Local closeness:** $\omega(S_v, S_{vu}) > \omega(S_v, S_{vu'})$ if $S_{vu}$ and $S_{vu'}$ are complete graphs with $S_{vu} = K_i$, $S_{vu'} = K_j$, and $i > j$, where $K_i$ refers to a complete graph on $i$ vertices.

(2) **Local denseness:** $\omega(S_v, S_{vu}) > \omega(S_v, S_{vu'})$ if $S_{vu}$ and $S_{vu'}$ have the same number of vertices but differ in the number of edges s.t. $|V_{vu}| = |V_{vu'}|$ and $|E_{vu}| > |E_{vu'}|$.

(3) **Isomorphic invariant:** $\omega(S_v, S_{vu}) = \omega(S_v, S_{vu'})$ if $S_{vu}$ and $S_{vu'}$ are isomorphic.

Figure 3 illustrates the first two properties. Let $\{\!\!\{\cdot\}\!\!\}$ denote a multiset, $\tilde{A} = (\tilde{A}_{vu})_{v,u \in V}$ where $\tilde{A}_{vu}$ is a normalised value of $A_{vu}$, and $X \in \mathbb{R}^{|V| \times f}$ be a matrix of input feature vectors where $x_v \in \mathbb{R}^f$ associates each $v \in V$. We denote the feature vector of $v$ at the t-th layer by $h_v^{(t)}$ and $h_v^{(0)} = x_v$. Then, the (t+1)-th layer of an aggregation scheme can be defined as:

$$m_a^{(t)} = \text{AGGREGATE}^N \left( \{\!\!\{ (\tilde{A}_{vu}, h_u^{(t)}) | u \in \mathcal{N}(v) \}\!\!\} \right), \tag{1}$$

$$m_v^{(t)} = \text{AGGREGATE}^I \left( \{\!\!\{ \tilde{A}_{vu} | u \in \mathcal{N}(v) \}\!\!\} \right) h_v^{(t)}, \tag{2}$$

$$h_v^{(t+1)} = \text{COMBINE} \left( m_v^{(t)}, m_a^{(t)} \right). \tag{3}$$

$\text{AGGREGATE}^N(\cdot)$ and $\text{AGGREGATE}^I(\cdot)$ are two possibly different parameterized functions. Here, $m_a^{(t)}$ is a message aggregated from the neighbors of $v$ and their structural coefficients, and $m_v^{(t)}$ is an

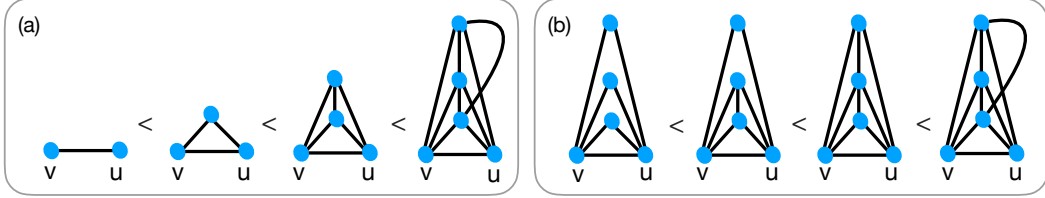

Figure 3: (a) *Local closeness*: for overlap subgraphs that are complete graphs, their structural coefficients increase with the number of vertices; (b) *Local denseness*: for overlap subgraphs that have the same number of vertices, their structural coefficients increase with the number of edges.

"adjusted" message from $v$ after performing an element-wise multiplication between $\text{AGGREGATE}^I(\cdot)$ and $h_v^{(t)}$ to account for structural effects from its neighbors. Then, $m_v^{(t)}$ and $m_a^{(t)}$ are combined by $\text{COMBINE}(\cdot)$ to obtain the feature vector $h_v^{(t+1)}$.

The following theorem states that a GNN can be more expressive than 1-WL if $\omega$ is powerful enough to distinguish structure beyond neighborhood subtrees and the neighborhood aggregation function $\Phi$ is injective under a sufficient number of layers. The proof is provided in Appendix C.

**Theorem 3.** *Let $M$ be a GNN whose aggregation scheme $\Phi$ is defined by Eq. 1-Eq. 3. $M$ is strictly more expressive than 1-WL in distinguishing non-isomorphic graphs if $M$ has a sufficient number of layers and also satisfies the following conditions:*

   *(1) $M$ can distinguish at least two neighborhood subgraphs $S_i$ and $S_j$ with $S_i \simeq_{subtree} S_j$, $S_i \not\simeq_{subgraph} S_j$ and $\{\!\{\tilde{A}_{iv'}|v' \in \mathcal{N}(i)\}\!\} \neq \{\!\{\tilde{A}_{ju'}|u' \in \mathcal{N}(j)\}\!\}$;*

   *(2) $\Phi\left(h_v^{(t)}, \{\!\{h_u^{(t)}|u \in \mathcal{N}(v)\}\!\}, \{\!\{(\tilde{A}_{vu}, h_u^{(t)})|u \in \mathcal{N}(v)\}\!\}\right)$ is injective.*

## 4 GRAPHSNN

Generally, there are many different ways of designing $\omega$ and $\Phi$ functions, leading to GNNs with different expressive powers. To elaborate this, we propose a novel GNN model, named GraphSNN, whose aggregation scheme is an instantiation of our generalised message-passing framework. We prove that the expressive power of GraphSNN goes beyond 1-WL.

**Model design.** In the following, we provide a definition of $\omega$ that satisfies the properties of local closeness, local denseness, and isomorphic invariant. One key idea behind this definition is to make it capable of being generalized to support different graph learning tasks, controlled by $\lambda > 0$ (will be further discussed in Section 5.3):

$$\omega(S_v, S_{vu}) = \frac{|E_{vu}|}{|V_{vu}| \cdot |V_{vu} - 1|}|V_{vu}|^\lambda. \tag{4}$$

This definition allows us to formulate a weighted adjacency matrix $A = (A_{vu})_{v,u \in V}$ for Graph-SNN. To compare structural coefficients across different nodes, we normalize $A$ to $\tilde{A}$ by $\tilde{A}_{vu} = \frac{A_{vu}}{\sum_{u \in \mathcal{N}(v)} A_{vu}}$. Alternatively, $A$ can be normalized using Softmax or other normalization techniques. For each vertex $v \in V$, the feature vector at the (t+1)-th layer is generated by

$$h_v^{(t+1)} = \text{MLP}_\theta\Big(\gamma^{(t)}\Big(\sum_{u \in \mathcal{N}(v)} \tilde{A}_{vu} + 1\Big)h_v^{(t)} + \sum_{u \in \mathcal{N}(v)}\Big(\tilde{A}_{vu} + 1\Big)h_u^{(t)}\Big), \tag{5}$$

where $\gamma^{(t)}$ is a learnable scalar parameter. Since $\mathcal{N}(v)$ refers to one-hop neighbors of $v$, one can stack multiple layers to handle more than one-hop neighborhood. Note that, to ensure the injectivity in the feature aggregation in the presence of structural coefficients, we add 1 into the first and second terms in Eq. 5. This design is critical for guaranteeing the expressiveness of GraphSNN beyond 1-WL, as will be discussed in the proofs of the lemmas and Theorem 4 later.

**Expressiveness analysis.** We first generalise the result of universal functions over *multisets* (Xu et al., 2019) to universal functions over *pairs of multisets* since Eq. 5 involves not only node features but

| Method | Cora | Citeseer | Pubmed | NELL | ogbn-arxiv |
|---|---|---|---|---|---|
| GCN | $81.5 \pm 0.4$ | $70.3 \pm 0.5$ | $79.0 \pm 0.5$ | $66.0 \pm 1.7$ | $71.74 \pm 0.29$ |
| GraphSNN$_{GCN}$ | $\textbf{83.1} \pm \textbf{1.8}$ | $\textbf{72.3} \pm \textbf{1.5}$ | $\textbf{79.8} \pm \textbf{1.2}$ | $\textbf{68.3} \pm \textbf{1.6}$ | $\textbf{72.20} \pm \textbf{0.90}$ |
| GAT | $83.0 \pm 0.6$ | $72.6 \pm 0.6$ | $78.5 \pm 0.3$ | - | - |
| GraphSNN$_{GAT}$ | $\textbf{83.8} \pm \textbf{1.2}$ | $\textbf{73.5} \pm \textbf{1.6}$ | $\textbf{79.6} \pm \textbf{1.4}$ | - | - |
| GIN | $77.6 \pm 1.1$ | $66.1 \pm 1.5$ | $77.0 \pm 1.2$ | $61.5 \pm 2.3$ | - |
| GraphSNN$_{GIN}$ | $\textbf{79.2} \pm \textbf{1.7}$ | $\textbf{68.3} \pm \textbf{1.5}$ | $\textbf{78.8} \pm \textbf{1.3}$ | $\textbf{63.8} \pm \textbf{2.7}$ | - |
| GraphSAGE | $79.2 \pm 3.7$ | $71.6 \pm 1.9$ | $77.4 \pm 2.2$ | $63.7 \pm 5.2$ | $71.49 \pm 0.27$ |
| GraphSNN$_{GraphSAGE}$ | $\textbf{80.5} \pm \textbf{2.5}$ | $\textbf{72.7} \pm \textbf{3.2}$ | $\textbf{79.0} \pm \textbf{3.5}$ | $\textbf{66.3} \pm \textbf{5.6}$ | $\textbf{71.80} \pm \textbf{0.70}$ |

Table 1: Classification accuracy (%) averaged over 10 runs on node classification.

also structural coefficients. Assume that $\mathcal{H}$, $\mathcal{A}$ and $\mathcal{W}$ are countable sets where $\mathcal{H}$ is a node feature space, $\mathcal{A}$ is a structural coefficient space, and $\mathcal{W} = \{A_{ij}h_i | A_{ij} \in \mathcal{A}, h_i \in \mathcal{H}\}$. Let $H$ and $W$ be two multisets containing elements from $\mathcal{H}$ and $\mathcal{W}$, respectively, and $|H| = |W|$. We can prove Lemma 1, Lemma 2 and Theorem 4 below, where the proof details are provided in Appendix C.

**Lemma 1.** *There exists a function $f$ s.t. $\pi(H, W) = \sum_{h \in H, w \in W} f(h, w)$ is unique for any distinct pair of multisets $(H, W)$.*

Then, the injectiveness of $\pi(H, W)$ can be extended to $\pi'(a, H, W)$ as in the lemma below.

**Lemma 2.** *There exists a function $f$ s.t. $\pi'(h_v, H, W) = \gamma f(h_v, |H|h_v) + \sum_{h \in H, w \in W} f(h, w)$ is unique for any distinct $(h_v, H, W)$, where $h_v \in \mathcal{H}$, $|H|h_v \in \mathcal{W}$, and $\gamma$ can be an irrational number.*

Since any function over $(h_v, H, W)$ can be decomposed as $g(\gamma f(h_v, |H|h_v) + \sum_{h \in H, w \in W} f(h, w))$, similar to Xu et al. (2019), we use a parameterized multi-layer perceptron (MLP) to learn $f$ and $g$. The following theorem characterises the expressive power of GraphSNN.

**Theorem 4.** *GraphSNN is more expressive than 1-WL in testing non-isomorphic graphs.*

Since GIN is as powerful as 1-WL (Xu et al., 2019), this theorem implies that GraphSNN is more expressive than GIN, i.e., GraphSNN can map at least two different neighborhood subgraphs that correspond to the same multiset of feature vectors to different representations.

**Complexity analysis.** Similar to GCN and GIN, GraphSNN is computationally efficient. The time complexity and memory complexity are linear w.r.t. the number of edges in a graph. Further, due to the locality of GraphSNN, the computation of aggregating feature vectors from neighborhood subgraphs at each layer can be parallelized across all vertices. Structural coefficients can be pre-computed with the time complexity $O(ml)$, where $m$ is the number of edges and $l$ is the maximum degree of vertices in a graph, and this computation can also be parallelized across all edges. Table 9 in Appendix A summarizes the time and space complexities of several popular message-passing GNNs in comparison with GraphSNN.

## 5 NUMERICAL EXPERIMENTS

In this section, we evaluate our models on node classification and graph classification benchmark tasks. All the results of our models are statistically significant at 0.05 level of significance.

### 5.1 NODE CLASSIFICATION

**Datasets.** We use five datasets: three citation network datasets Cora, Citeseer, and Pubmed (Sen et al., 2008) for semi-supervised document classification, one knowledge graph dataset NELL (Carlson et al., 2010) for semi-supervised entity classification, and one OGB dataset ogbn-arxiv from (Hu et al., 2020). Table 10 in Appendix B contains statistics for these datasets.

**Baseline methods.** We consider the popular message-passing GNNs: GCN (Kipf & Welling, 2017), GAT (Veličković et al., 2017), GIN (Xu et al., 2019), and GraphSAGE (Hamilton et al., 2017). For each of these baselines, we construct a GraphSNN$_M$ model by replacing its aggregation scheme by our aggregation scheme, which is detailed in Appendix A. The purpose of this setup is to evaluate

| Method | MUTAG | PTC-MR | PROTEINS | D&D | BZR | COX2 | IMDB-B | RDT-M5K |
|---|---|---|---|---|---|---|---|---|
| WL | 90.4 ± 5.7 | 59.9 ± 4.3 | 75.0 ± 3.1 | 79.4 ± 0.3 | 78.5 ± 0.6 | 81.7 ± 0.7 | 73.8 ± 3.9 | 52.5 ± 2.1 |
| RetGK | 90.3 ± 1.1 | 62.5 ± 1.6 | 75.8 ± 0.6 | 81.6 ± 0.3 | - | - | 71.9 ± 1.0 | - |
| GNTK | 90.0 ± 8.5 | **67.9 ± 6.9** | 75.6 ± 4.2 | 75.6 ± 3.9 | 83.6 ± 2.9 | - | 76.9 ± 3.6 | - |
| P-WL | 90.5 ± 1.3 | 64.0 ± 0.8 | 75.2 ± 0.3 | 78.6 ± 0.3 | - | - | - | - |
| WL-PM | 87.7 ± 0.8 | 61.4 ± 0.8 | - | 78.6 ± 0.2 | - | - | - | - |
| WWL | 87.2 ± 1.5 | 66.3 ± 1.2 | 74.2 ± 0.5 | 79.6 ± 0.5 | 84.4 ± 2.0 | 78.2 ± 0.4 | 74.3 ± 0.8 | - |
| FGW | 88.4 ± 5.6 | 65.3 ± 7.9 | 74.5 ± 2.7 | - | 85.1 ± 4.1 | 77.2 ± 4.8 | 63.8 ± 3.4 | - |
| DGCNN | 85.8 ± 1.7 | 58.6 ± 2.5 | 75.5 ± 0.9 | 79.3 ± 0.9 | - | - | 70.0 ± 0.9 | 48.7 ± 4.5 |
| CapsGNN | 86.6 ± 6.8 | 66.0 ± 1.8 | 76.2 ± 3.6 | 75.4 ± 4.1 | - | - | 73.1 ± 4.8 | 52.9 ± 1.5 |
| [†]GraphSAGE | 85.1 ± 7.6 | 63.9 ± 7.7 | 75.9 ± 3.2 | 72.9 ± 2.0 | - | - | 72.3 ± 5.3 | 50.0 ± 1.3 |
| [†]GIN | 89.4 ± 5.6 | 64.6 ± 7.0 | 75.9 ± 2.8 | - | - | - | 75.1 ± 5.1 | 57.5 ± 1.5 |
| [†]GraphSNN (S) | **91.57 ± 2.8** | 66.70 ± 3.7 | **76.83 ± 2.5** | 81.97 ± 2.6 | 88.69 ± 3.2 | 82.86 ± 3.1 | 77.86 ± 3.6 | 58.43 ± 2.3 |
| [†]GraphSNN (R) | **91.24 ± 2.5** | 66.96 ± 3.5 | 76.51 ± 2.5 | 82.46 ± 2.7 | 88.97 ± 2.9 | 83.13 ± 3.5 | 76.93 ± 3.3 | 58.51 ± 2.7 |
| GraphSNN (S) | **94.70 ± 1.9** | 70.58 ± 3.1 | 78.42 ± 2.7 | 83.92 ± 2.3 | 91.12 ± 3.0 | 86.28 ± 3.3 | 78.51 ± 2.8 | 59.86 ± 2.6 |
| GraphSNN (R) | **94.14 ± 1.2** | 71.01 ± 3.6 | 78.21 ± 2.9 | 84.61 ± 1.5 | 91.88 ± 3.2 | 86.72 ± 2.9 | 77.87 ± 3.1 | 60.23 ± 2.2 |

Table 2: Classification accuracy (%) averaged over 10 runs on graph classification. The results of WL and RetGK are taken from (Du et al., 2019), GraphSAGE from (Xu et al., 2019), DGCNN from (Maron et al., 2019) and others from their original papers. † indicates the reporting setting used in GIN and further details on the experimental settings are discussed in Appendix B.

how effectively our aggregation scheme with structural coefficients can learn representations for vertices, compared with the standard message-passing aggregation scheme.

**Experimental setup.** We use the Adam optimizer (Kingma & Ba, 2015) and $\lambda = 1$. For ogbn-arxiv, our models are trained for 500 epochs with the learning rate 0.01, dropout 0.5, hidden units 256, and $\gamma = 0.1$. For the other datasets, we use 200 epochs with the learning rate 0.001, and choose the best values for weight decay from $\{0.001, 0.002, ..., 0.009\}$ and hidden units from $\{64, 128, 256, 512\}$. For $\gamma$ and dropout at each layer, the best value for each model in each dataset is selected from $\{0.1, 0.2, ..., 0.6\}$. GraphSNN$_{GAT}$ uses the attention dropout 0.6 and 8 multi-attention heads. GraphSNN$_{GraphSAGE}$ uses the neighborhood sample size 25 with the mean aggregation.

We consider two settings of data splits for all datasets except for ogbn-arxiv: (1) the standard splits in Kipf & Welling (2017), i.e., 20 nodes from each class for training, 500 nodes for validation and 1000 nodes for testing, for which the results are presented in Table 1; (2) the random splits in Pei et al. (2020), i.e., randomly splitting nodes into 60%, 20% and 20% for training, validation and testing, respectively, for which the results are presented in Table 13 in Appendix B. For ogbn-arxiv, we follow Hu et al. (2020) to use a time-based data split based on publication dates.

## 5.2 GRAPH CLASSIFICATION

We evaluate GraphSNN from three aspects: (1) small standard graph datasets, (2) large graph datasets and (3) comparison with GNNs that are go beyond 1-WL.

**Experiments on small graphs.** We use eight datasets from two categories: (1) bioinformatics datasets: MUTAG, PTC-MR, COX2, BZR, PROTEINS, and D&D (Debnath et al., 1991; Kriege et al., 2016; Wale et al., 2008; Shervashidze et al., 2011; Sutherland et al., 2003; Borgwardt & Kriegel, 2005); (2) social network datasets: IMDB-B and RDT-M5K (Yanardag & Vishwanathan, 2015). Table 11 in Appendix B contains statistics for these small graph datasets.

We compare against eleven baselines: (1) *Graph kernel based methods:* WL subtree kernel (Shervashidze et al., 2011), RetGK (Zhang et al., 2018b), GNTK (Du et al., 2019), P-WL (Rieck et al., 2019), WL-PM (Nikolentzos et al., 2017), WWL (Togninalli et al., 2019) and FGW (Titouan et al., 2019); (2) *GNN based methods:* DGCNN (Zhang et al., 2018a), CapsGNN (Xinyi & Chen, 2018), GIN (Xu et al., 2019), and GraphSAGE (Hamilton et al., 2017).

Both the standard stratified splits (Xu et al., 2019) and the random splits are considered. We use 10-fold cross validation with 90% training and 10 % testing, and report the best mean accuracy. For both settings, we use the Adam optimizer (Kingma & Ba, 2015), batch size 64, hidden dimension 64, weight decay of 0.009, a 2-layer MLP with batch normalization, 500 epochs and dropout of 0.6, and $\gamma = 0.1$ over all datasets. The readout function as in (Xu et al., 2019) is used which concatenates representations of all layers to obtain a final graph representation. For the standard stratified splits, we use the learning rate 0.009 over all datasets. For the random splits, we use the learning rate 0.008 for MUTAG and RDT-M5K, and 0.007 for the other datasets. Table 2 presents the results.

| Method | ogbg-molhiv | ogbg-moltox21 | ogbg-moltoxcast | ogbg-ppa | ogbg-molpcba |
|---|---|---|---|---|---|
| GIN | 75.58±1.40 | 74.91±0.51 | 63.41±0.74 | 68.92±1.00 | 22.66±0.28 |
| GIN+VN | 75.20±1.30 | 76.21±0.82 | 66.18±0.68 | 70.37±1.07 | 27.03±0.23 |
| GSN | 77.99±1.00 | - | - | - | - |
| PNA | 79.05±1.30 | - | - | - | 28.38±0.35 |
| ID-GNN | 78.30±2.00 | - | - | - | - |
| Deep LRP | 77.19±1.40 | - | - | - | - |
| GraphSNN | 78.51±1.70 | 75.45±1.10 | 65.40±0.71 | 70.66±1.65 | 24.96±1.50 |
| GraphSNN+VN | **79.72±1.83** | **76.78±1.27** | **67.68±0.92** | **72.02±1.48** | **28.50±1.68** |

Table 3: Classification accuracy (%) averaged over 10 runs on graph classification, where $\lambda = 2$. The results of the baselines are taken from (Hu et al., 2020) and the leaderboard of the OGB website.

| | Method | MUTAG | PTC-MR | PROTEINS | BZR | IMDB-B |
|---|---|---|---|---|---|---|
| GSN | GSN-e | 90.6 ± 7.5 | **68.2 ± 7.2** | 76.6 ± 5.0 | - | 77.8 ± 3.3 |
| | GSN-v | 92.2 ± 7.5 | 67.4 ± 5.7 | 74.5 ± 5.0 | - | 76.8 ± 2.0 |
| ID-GNNs | ID-GNN Fast | **96.5 ± 3.2** | 61.9 ± 5.4 | **78.0 ± 3.5** | 86.4 ± 3.0 | - |
| | ID-GNN Full | 93.0 ± 5.6 | 62.5 ± 5.3 | 77.9 ± 2.4 | 88.1 ± 4.0 | - |
| Ours | GraphSNN | 91.57 ± 2.8 | 66.70 ± 3.7 | 76.83 ± 2.5 | **88.69 ± 3.2** | **77.86 ± 3.6** |
| k-WL GNNs | 1-GNN$_{NT}$ | 82.7 ± 0.0 | 51.2 ± 0.0 | - | - | 69.4 ± 0.0 |
| | 1-GNN | 82.2 ± 0.0 | 59.0 ± 0.0 | - | - | 71.2 ± 0.0 |
| | 1-2-3-GNN$_{NT}$ | 84.4 ± 0.0 | 59.3 ± 0.0 | - | - | 70.3 ± 0.0 |
| | 1-2-3-GNN | 86.1 ± 0.0 | 60.9 ± 0.0 | - | - | 74.2 ± 0.0 |
| Ours | GraphSNN | **87.30 ± 3.1** | 61.63 ± 2.8 | 74.01 ± 3.2 | 82.72 ± 3.9 | 74.81 ± 3.5 |

Table 4: Classification accuracy (%) averaged over 10 runs on graph classification, where $\lambda = 2$. The results of the baselines are taken from their original papers. GSN and ID-GNNs use the same experimental setup as GIN, while k-WL GNNs uses the same experimental setup as CapsGNN. These experimental setups are detailed in Appendix B.

**Experiments on large graphs.** We use five large graph datasets from Open Graph Benchmark (OGB) Hu et al. (2020), including four molecular graph datasets (ogbg-molhiv, ogbg-moltox21, ogbg-moltoxcast and ogb-molpcba) and one protein-protein association network (ogbg-ppa). Table 12 in Appendix B contains statistics for these large graph datasets.

We compare against the following methods that have reported the results on the above OGB datasets: GIN and GIN+VN (Hu et al., 2020), GSN (Bouritsas et al., 2020), PNA (Corso et al., 2020), ID-GNNs (You et al., 2021) and Deep LRP (Chen et al., 2020b). In addition to the original model of GraphSNN, we also consider a variant, denoted as GraphSNN+VN, which performs the message passing over augmented graphs with virtual nodes in GraphSNN (Hu et al., 2020; Ishiguro et al., 2019).

We follow the same experiment setup as in Hu et al. (2020). We use the Adam optimizer with learning rate 0.001, batch size 32, dropout 0.5 and 100 epochs for all datasets. GraphSNN uses a 8-layer MLP with embedding dimension 512 for ogbg-moltoxcast and ogbg-moltox21, while GraphSNN+VN has the embedding dimensions 300 and 256, and 8-layer and 5-layer MLPs for ogbg-moltoxcast and ogbg-moltox21, respectively. For ogbg-molhiv, ogbg-molpcba and ogbg-ppa, both GraphSNN and GraphSNN+VN use a 5-layer MLP and embedding dimension 200. Table 3 shows the results for the classification accuracy. Table 15 in Appendix B shows the results for the running time of the prepocessing step.

**Comparison with GNNs beyond 1-WL.** We compare GraphSNN with the other GNNs that are more expressive than 1-WL, including: GSN (Bouritsas et al., 2020), ID-GNNs (You et al., 2021) and k-WL GNN (Morris et al., 2019). We use the same experimental setup as in (Xu et al., 2019; Bouritsas et al., 2020; Maron et al., 2019). Table 4 shows the results.

## 5.3 ABLATION STUDY

We perform an ablation study to analyze the effect of $\lambda$ values on model performance. Tables 5 and 6 show that $\lambda = 1$ yields the highest performance for node classification, while $\lambda = 2$ is the best for graph classification. This reflects a critical point - different classes of structure information are needed by different graph learning tasks. $\lambda = 1$ captures local density, e.g., two overlap subgraphs may

| Dataset | Method | $\lambda=1$ | $\lambda=2$ | $\lambda=3$ | $\lambda=4$ | $\lambda=5$ |
|---|---|---|---|---|---|---|
| Cora | GraphSNN$_{GCN}$ | **83.1±1.8** | 82.8±1.3 | 82.3±2.4 | 81.8±1.6 | 82.1±1.6 |
| | GraphSNN$_{GIN}$ | **79.2±1.7** | 78.8±1.2 | 78.5±1.3 | 78.1±1.6 | 77.7±1.2 |
| | GraphSNN$_{GraphSAGE}$ | **80.5±2.5** | 80.3±2.1 | 79.8±1.9 | 79.2±1.9 | 79.4±2.2 |
| | GraphSNN$_{GAT}$ | **83.8±1.2** | 83.5±1.5 | 83.2±1.7 | 82.8±1.3 | 83.2±1.9 |
| Citeseer | GraphSNN$_{GCN}$ | **72.3±1.5** | 71.7±1.3 | 71.1±1.6 | 70.6±1.2 | 70.9±1.1 |
| | GraphSNN$_{GIN}$ | **68.3±1.5** | 68.3±1.9 | 67.7±1.4 | 67.1±1.3 | 67.3±1.4 |
| | GraphSNN$_{GraphSAGE}$ | **72.7±3.2** | 72.0±2.5 | 71.6±2.9 | 71.9±2.1 | 71.3±2.3 |
| | GraphSNN$_{GAT}$ | **73.5±1.6** | 72.9±1.7 | 72.5±1.1 | 72.6±1.6 | 72.0±1.3 |

Table 5: Classification accuracy (%) averaged over 10 runs on node classification with standard splits.

| Dataset | Method | $\lambda=1$ | $\lambda=2$ | $\lambda=3$ | $\lambda=4$ | $\lambda=5$ |
|---|---|---|---|---|---|---|
| MUTAG | | 92.66±2.4 | **94.14±1.2** | 93.38±1.5 | 92.25±2.1 | 92.79±2.0 |
| PTC-MR | | 70.76±5.1 | **71.01±3.6** | 70.67±2.8 | 69.59±2.1 | 69.97±3.1 |
| PROTEINS | | 77.90±4.9 | **78.21±2.9** | 78.15±2.1 | 77.20±3.1 | 76.93±3.2 |
| D&D | GraphSNN | 82.70±4.6 | **84.61±1.5** | 84.34±1.2 | 82.60±2.6 | 82.30±2.3 |
| BZR | | 87.61±4.9 | **91.88±3.2** | 91.45±2.6 | 91.38±2.1 | 90.90±3.1 |
| COX2 | | 86.20±3.3 | **86.72±2.9** | 83.81±3.1 | 83.13±2.6 | 83.94±3.2 |
| IMDB-B | | 77.07±5.2 | **77.87±3.1** | 77.60±3.6 | 77.32±3.2 | 77.10±3.3 |
| RDT-M5K | | 59.53±2.6 | **60.23±2.2** | 60.10±2.3 | 60.00±2.1 | 59.90±2.6 |

Table 6: Classification accuracy (%) averaged over 10 runs on graph classification with random splits.

considerably vary in the number of vertices but their local density can be very close. Our experiments show that injecting such local density helps improve the performance of node classification. $\lambda = 2$ captures local similarity, i.e., how similar two overlap subgraphs are. Two overlap subgraphs that considerably differ in the number of vertices would have very different structural coefficients. Since graph classification requires to compare the similarity of two graphs, $\lambda = 2$ is thus the best.

## 5.4 OVERSMOOTHING ANALYSIS

We analyse the impact of model depth (number of layers) on node classification performance. In addition to GCN and GraphSNN$_{GCN}$, we also compare these models with a residual connection (i.e., GCN+residual and GraphSNN$_{GCN}$+residual). We evaluate all the models on Cora dataset using the standard splits and same hyperparameters as in Section 5.1. Table 7 shows the results. When increasing the model depth, GraphSNN$_{GCN}$ performs consistently better than GCN at each layer. This is because structural coefficients capture structural connectivity between a target vertex and its neighbors. Thus, a neighbor whose structural connectivity is weak would pass little messages to the target vertex, whereas a neighbor whose structural connectivity is strong would pass a strong message to the target vertex. GraphSNN helps alleviate the oversmoothing issue even in the presence of residual connections. Further results of the oversmoothing analysis are provided in Appendix B.

| #Layers | GCN | GCN+residual | GraphSNN$_{GCN}$ | GraphSNN$_{GCN}$+residual |
|---|---|---|---|---|
| 1 | 79.6±0.5 | 80.3±0.7 | 80.1±0.8 | 81.6±1.6 |
| 2 | **81.5±0.4** | **82.8±1.2** | **83.1±1.8** | **84.1±1.7** |
| 3 | 80.3±0.6 | 82.3±0.5 | 82.0±0.8 | 83.4±0.7 |
| 4 | 78.2±0.9 | 81.5±0.9 | 80.1±0.7 | 82.9±0.9 |
| 5 | 74.3±1.3 | 81.0±1.3 | 79.1±1.2 | 82.3±0.3 |
| 6 | 35.6±1.5 | 80.6±0.5 | 76.5±1.3 | 81.5±1.2 |
| 7 | 31.6±0.9 | 79.7±0.6 | 76.3±1.3 | 80.9±0.9 |
| 8 | 16.2±1.2 | 78.4±1.1 | 75.7±1.2 | 80.3±1.3 |

Table 7: Classification accuracy (%) averaged over 10 runs on Cora dataset.

## 6 CONCLUSIONS

In this paper, we have introduced a GNN framework, which enables a general way of injecting structural information into a message-passing aggregation scheme. We have also introduced a novel GNN model, GraphSNN, for graph learning, and prove that GraphSNN is more expressive than 1-WL in distinguishing graph structures. It is shown that GraphSNN consistently outperforms all the state-of-the-art approaches in both node classification and graph classification benchmark tasks.

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

APPENDIX

## A. CONNECTIONS TO PREVIOUS WORK

In the following, we discuss how our framework generalizes the existing message-passing GNNs in the literature such as GCN (Kipf & Welling, 2017), GraphSAGE (Hamilton et al., 2017), GAT (Veličković et al., 2017) and GIN (Xu et al., 2019) as special cases. Table 8 presents the local aggregation schemes used by these existing GNN models. They differ from each other w.r.t. the way of aggregating feature vectors in a neighborhood and how they are combined with the current vertex's feature itself, i.e., summation or concatenation. Here, $\alpha_{vu}$ is an attention coefficient capturing the importance of a neighbor in GAT, $\epsilon$ is a learnable or fixed scalar parameter used in GIN, $W$ is a learnable weight matrix and $\sigma$ is a non-linear activation function, such as ReLU.

Note that, as defined in Equation 3, $m_a^{(t)}$ and $m_v^{(t)}$ refer to the messages aggregated by AGGREGATE$^N(\cdot)$ and AGGREGATE$^I(\cdot)$, respectively.

| GNN Model | AGGREGATE$^N(\cdot)$ | AGGREGATE$^I(\cdot)$ | COMBINE$(\cdot)$ |
|---|---|---|---|
| GCN | $\sum_{u \in \mathcal{N}(v)} \frac{W^{(t)} h_u^{(t)}}{\sqrt{|\mathcal{N}(u)||\mathcal{N}(v)|}}$ | $\frac{W^{(t)} h_v^{(t)}}{\sqrt{|\mathcal{N}(v)||\mathcal{N}(v)|}}$ | $\sigma(\text{SUM}(m_v^{(t)}, m_a^{(t)}))$ |
| GraphSAGE | $\sum_{u \in \mathcal{N}(v)} \frac{h_u^{(t)}}{|\mathcal{N}(v)|}$ | $h_v^{(t)}$ | $\sigma(W^{(t)} \cdot \text{CONCAT}(m_v^{(t)}, m_a^{(t)}))$ |
| GAT | $\sum_{u \in \mathcal{N}(v)} \alpha_{vu} W^{(t)} h_u^{(t)}$ | $\alpha_{vv} W^{(t)} h_v^{(t)}$ | $\sigma(\text{SUM}(m_v^{(t)}, m_a^{(t)}))$ |
| GIN | $\sum_{u \in \mathcal{N}(v)} h_u^{(t)}$ | $(1 + \epsilon) h_v^{(t)}$ | $\text{MLP}_\theta(\text{SUM}(m_v^{(t)}, m_a^{(t)}))$ |

Table 8: Comparison of the aggregation schemes used in existing message-passing GNNs

COMPLEXITY ANALYSIS

Table 9 summarizes the time and space complexities of several popular message-passing GNNs and GraphSNN, where $n$ and $m$ are the numbers of vertices and edges in a graph, respectively, $k$ refers to the number of layers, $f$ and $d$ are the dimensions of input and output feature vectors, respectively, $a$ is the number of attention heads used in GAT, and $s$ is the number of neighbors sampled for each node at each layer in GraphSAGE.

| GNN Model | Time Complexity | Memory Complexity |
|---|---|---|
| GCN (Kipf & Welling, 2017) | $O(kmfd)$ | $O(m)$ |
| GIN (Xu et al., 2019) | $O(kmfd)$ | $O(m)$ |
| GAT (Veličković et al., 2017) | $O(k(anfd + amd))$ | $O(n^2)$ |
| GraphSAGE (Hamilton et al., 2017) | $O(snfd)$ | $O(n)$ |
| GraphSNN (ours) | $O(kmfd)$ | $O(m)$ |

Table 9: Time and space complexities of message-passing GNNs and GraphSNN.

FORMULATION OF GRAPHGNN$_M$

For each of these message-passing GNNs, denoted as $M$, we construct a variant GraphSNN$_M$ by replacing its existing aggregation scheme by our aggregation scheme with structural coefficients as formulated in Eq. 5. These variants are used in our experiments for node classification benchmark tasks (see Section 5.1) in order to evaluate how our aggregation scheme with structural coefficients can improve performance, compared with their standard message-passing aggregation schemes. Below are the details of these variants.

**GCN and GraphSNN$_{GCN}$**

Graph Convolutional Network (GCN) (Kipf & Welling, 2017) applies a normalized mean aggregation to combine the feature vector of a node $v$ with the feature vectors in its neighborhood $\mathcal{N}(v)$:

$$h_v^{(t+1)} = \sigma\Big(\frac{W^{(t)}h_v^{(t)}}{\sqrt{|\mathcal{N}(v)||\mathcal{N}(v)|}} + \sum_{u\in\{\mathcal{N}(v)\}}\frac{W^{(t)}h_u^{(t)}}{\sqrt{|\mathcal{N}(v)||\mathcal{N}(u)|}}\Big). \tag{6}$$

$\sqrt{|\mathcal{N}(u)||\mathcal{N}(v)|}$ is a normalization constant for the edge $(v,u)$, which originates from the normalized adjacency matrix $D^{-1/2}AD^{-1/2}$. $W^{(t)}$ is a trainable weight matrix and $\sigma$ is a non-linear activation function such as ReLU. We generalise GCN to a model under the GMP framework, namely **GraphSNN**$_{GCN}$, to improve the expressive power of GCN. We first construct a normalized structural coefficient matrix $\tilde{A}$. Formally, each neural layer of GraphSNN$_{GCN}$ may then be expressed as:

$$h_v^{(t+1)} = \sigma\Bigg(\gamma^{(t)}\Big(\sum_{u\in\mathcal{N}(v)}\tilde{A}_{vu}+1\Big)\frac{W^{(t)}h_v^{(t)}}{\sqrt{|\tilde{\mathcal{N}}(v)||\tilde{\mathcal{N}}(v)|}} + \sum_{u\in\mathcal{N}(v)}\Big(\tilde{A}_{vu}+1\Big)\frac{W^{(t)}h_u^{(t)}}{\sqrt{|\tilde{\mathcal{N}}(u)||\tilde{\mathcal{N}}(v)|}}\Bigg). \tag{7}$$

**GraphSAGE and GraphSNN$_{GraphSAGE}$**

GraphSAGE (Hamilton et al., 2017) learns aggregation functions to induce new node feature vectors by sampling and aggregating features from a node's local neighborhood. GraphSAGE has considered three different aggregation functions such as mean aggregator, LSTM aggregator and pooling aggregator. In our work, we mainly focus on the mean aggregator that, for each vertex $v$, takes the mean of the feature vectors of the nodes in its neighborhood and concatenates it with the feature vector of $v$ as shown below:

$$h_v^{(t+1)} = \sigma\Big(W^{(t)}\cdot\text{CONCAT}\Big(\frac{1}{|\mathcal{N}(v)|}\sum_{u\in\mathcal{N}(v)}h_u^{(t)}, h_v^{(t)}\Big)\Big), \tag{8}$$

where $W^{(t)}$ is a learnable weight matrix, and $\sigma$ represents a non-linear activation function. We also generalise GraphSNN to a model under the GMP framework, namely **GraphSNN$_{GraphSAGE}$**. This model first takes a mean aggregation of the feature vectors in the neighborhood $\mathcal{N}(v)$ and then concatenates it with the feature vector of $v$ itself in the following manner:

$$h_v^{(t+1)} = \sigma\Big(W^{(t)}\cdot\text{CONCAT}\Big(\frac{1}{|\mathcal{N}(v)|}\sum_{u\in\mathcal{N}(v)}\Big(\tilde{A}_{vu}+1\Big)h_u^{(t)}, \gamma^{(t)}\Big(\sum_{u\in\mathcal{N}(v)}\tilde{A}_{vu}+1\Big)h_v^{(t)}\Big)\Big). \tag{9}$$

**GAT and GraphSNN$_{GAT}$**

Graph Attention Network (GAT) (Veličković et al., 2017) linearly transforms the input feature vectors and performs a weighted sum of the feature vectors for vertices in a neighborhood after the transformation. GAT computes attention weights $\alpha_{vu}^{(t)}$ using an attention mechanism and aggregates the feature vectors in a neighborhood as follows:

$$h_v^{(t+1)} = \sigma\Big(\sum_{(v,u)\in E}\alpha_{vu}^{(t)}W^{(t)}h_u^{(t)}\Big), \tag{10}$$

where $W^{(t)}$ is a trainable weight matrix and $\sigma$ represents a non-linear activation function. We generalise GAT to a model, called **GraphSNN$_{GAT}$**, in the GMP framework. Firstly, we aggregate the feature vectors based on structural coefficients in our aggregation scheme, i.e., we compute

$$\tilde{h}_u^{(t)} = \gamma^{(t)}\Big(\sum_{z\in\mathcal{N}(u)}\tilde{A}_{uz}+1\Big)\frac{h_u^{(t)}}{\sqrt{|\tilde{\mathcal{N}}(u)||\tilde{\mathcal{N}}(u)|}} + \sum_{z\in\mathcal{N}(u)}\Big(\tilde{A}_{uz}+1\Big)\frac{h_z^{(t)}}{\sqrt{|\tilde{\mathcal{N}}(z)||\tilde{\mathcal{N}}(u)|}} \tag{11}$$

and

$$\tilde{h}_v^{(t)} = \gamma^{(t)}\Big(\sum_{z'\in\mathcal{N}(v)}\tilde{A}_{vz'}+1\Big)\frac{h_v^{(t)}}{\sqrt{|\tilde{\mathcal{N}}(v)||\tilde{\mathcal{N}}(v)|}} + \sum_{z'\in\mathcal{N}(v)}\Big(\tilde{A}_{vz'}+1\Big)\frac{h_{z'}^{(t)}}{\sqrt{|\tilde{\mathcal{N}}(z')||\tilde{\mathcal{N}}(v)|}}. \tag{12}$$

We then construct attention coefficients $\alpha_{vu}^{(t)}$ on these aggregated feature vectors as follows:

$$\alpha_{vu}^{(t)} = \frac{\exp\Big(\text{LeakyReLU}\big(a^T[W^{(t)}\tilde{h}_v^{(t)}||W^{(t)}\tilde{h}_u^{(t)}]\big)\Big)}{\sum_{z\in\mathcal{N}(v)}\exp\Big(\text{LeakyReLU}\big(a^T[W^{(t)}\tilde{h}_v^{(t)}||W^{(t)}\tilde{h}_z^{(t)}]\big)\Big)}, \tag{13}$$

where $||$ represents the concatenation, $W^{(t)}$ is a learnabe weight matrix and $a$ is a learnable weight vector. After that, we aggregate the neighborhood features as follows using attention coefficients.

$$h_v^{(t+1)} = \sigma\Big(\sum_{(v,u)\in E}\alpha_{vu}^{(t)}W^{(t)}\tilde{h}_u^{(t)}\Big), \tag{14}$$

where $W^{(t)}$ is a learnable weight matrix, and $\sigma$ represents a non-linear activation function. We use multi-head attention as stated in the original work Veličković et al. (2017).

**GIN and GraphSNN$_{GIN}$**

Graph Isomorphism Network (GIN) (Xu et al., 2019) takes the sum aggregation over a neighborhood, followed by a 2-layer MLP. The $\epsilon^{(t+1)}$ is a learnable parameter or fixed scalar. Each neural layer is expressed as:

$$h_v^{(t+1)} = \text{MLP}^{(t+1)}\Big((1+\epsilon^{(t+1)})h_v^{(t)} + \sum_{u\in\mathcal{N}(v)}h_u^{(t)}\Big). \tag{15}$$

Here, we consider one of GIN variants employed in the original paper, where the learnable parameter $\epsilon = 0$, and generalise it to **GraphSNN$_{GIN}$** as defined bwlow:

$$h_v^{(t+1)} = \text{MLP}^{(t+1)}\Big(\gamma^{(t)}\Big(\sum_{u\in\mathcal{N}(v)}\tilde{A}_{vu}+1\Big)h_v^{(t)} + \sum_{u\in\mathcal{N}(v)}\Big(\tilde{A}_{vu}+1\Big)h_u^{(t)}\Big). \tag{16}$$

## B. EXPERIMENTS

### DATASETS

Table 10 contains the statistics for the five datasets used in our experiments for node classification in Section 5.1.

| Dataset | Type | #Nodes | #Edges | #Classes | #Features |
|---------|------|--------|--------|----------|-----------|
| Cora | Citation network | 2,708 | 5,429 | 7 | 1,433 |
| Citeseer | Citation network | 3,327 | 4,732 | 6 | 3,703 |
| Pubmed | Citation network | 19,717 | 44,338 | 3 | 500 |
| NELL | Knowledge graph | 65,755 | 266,144 | 210 | 5,414 |
| ogbn-arxiv | Citation network | 169,343 | 1,166,243 | 40 | 128 |

Table 10: Statistics for node classification datasets.

Table 11 below contains the statistics for the datasets used in our experiments on small graph classification in Section 5.2, as well as the datasets used in an additional experiment for graph classification following the data splits and experimental setup in (Errica et al., 2020). The results of this additional experiment are reported under Section "Graph Classification using Setup (Errica et al., 2020)" in Appendix B.

Table 12 contains the statistics for the five large graph datasets from from Open Graph Benchmark (OGB) Hu et al. (2020), used in our experiments for large graph classification in Section 5.2.

### EXPERIMENTAL SETUP ON SMALL GRAPHS

Previously, several experimental setups have been considered for evaluating graph classification on small graphs in TUD benchmark datasets (https://chrsmrrs.github.io/datasets/). All the baseline methods in our paper use the 10-fold cross validation technique. However, they differ in how they split training/validation/testing data and how they report the final results in terms of classification accuracy. Below, we discuss the details of their experimental setups.

| Dataset | #Graphs | Avg # Nodes | Avg # Edges | #Classes |
|---------|---------|-------------|-------------|----------|
| MUTAG | 188 | 17.93 | 19.79 | 2 |
| PTC-MR | 344 | 14.29 | 14.69 | 2 |
| BZR | 405 | 35.75 | 38.36 | 2 |
| COX2 | 467 | 41.22 | 43.45 | 2 |
| ENZYMES | 600 | 32.63 | 64.14 | 6 |
| IMDB-B | 1000 | 19.77 | 96.53 | 2 |
| PROTEINS | 1113 | 39.06 | 72.82 | 2 |
| D & D | 1178 | 284.32 | 715.66 | 2 |
| NCI1 | 4110 | 29.87 | 32.30 | 2 |
| RDT-M5K | 5000 | 508.52 | 594.87 | 5 |
| COLLAB | 5000 | 74.49 | 2457.78 | 3 |

Table 11: Statistics for small graph classification datasets.

| Dataset | #Graphs | Avg # Nodes | Avg # Edges | #Tasks | Task Type |
|---------|---------|-------------|-------------|--------|-----------|
| ogbg-molmolhiv | 41,127 | 25.5 | 27.5 | 1 | Binary classification |
| ogbg-moltox21 | 7,831 | 18.6 | 19.3 | 12 | Binary classification |
| ogbg-moltoxcast | 8,576 | 18.8 | 19.3 | 617 | Binary classification |
| ogbg-molpcba | 437,929 | 26.0 | 28.1 | 128 | Binary classification |
| ogbg-ppa | 158,100 | 243.4 | 2,266.1 | 1 | Multi-class classification |

Table 12: Statistics for large graph classification dataset (OGB graph datasets).

- CapsGNN (Xinyi & Chen, 2018) splits the datasets into 80 % for training, 10 % for validation, and 10 % for testing. The training is stopped when the performance on the validation set goes to the highest. Then they obtain the test set accuracy that corresponds to the epoch with the highest validation accuracy in each fold. The final results are reported by computing the mean accuracy and standard deviation over 10 folds.

- DGCNN Zhang et al. (2018a) splits the datasets into 90 % for training and 10 % for testing. They obtain the test accuracy of the last epoch in each fold. They report the final results by computing the mean accuracy and standard deviation on the test accuracy over 10 folds.

- GIN and GraphSAGE (Xu et al., 2019) split the datasets into 90 % for training and 10 % for testing. They average the test accuracy on 10 folds and select the epoch with the highest averaged accuracy. Then they report the final results by computing the mean accuracy and standard deviation based on the selected epoch.

- FGW (Titouan et al., 2019) splits the datasets into 90 % for training and 10 % for testing. Then, they use the nested cross validation technique on the same folds, and repeat the process 10 times. They report the final results by computing the mean accuracy and standard deviation.

- The other baseline methods split the datasets into 90 % for training and 10 % for testing, and repeat their experiment 10 times. Then they report the final results by computing the mean accuracy and standard deviation.

In our work, we split the datasets into 90 % for training and 10 % for testing. We obtain the best validation accuracy on each fold. Then we report the final results by computing the mean accuracy and standard deviation over 10 folds[1].

NODE CLASSIFICATION USING RANDOM SPLITS

Following the work Pei et al. (2020), we randomly split graph nodes into 60%, 20% and 20% for training, validation and testing, respectively. The other hyperparameter settings are the same as in Section 5.1. Table 13 shows the results. We see that our models consistently outperform all of the baseline methods on all benchmark datasets. Specifically, GraphSN$_{GCN}$ improves upon GCN by a margin of 1.5%, 1.7%, 1.6% and 2.4% on Cora, Citeseer, Pubmed and NELL, respectively.

---

[1]The implementation can be found at: https://github.com/wokas36/GraphSNN

GraphSN$_{GAT}$ improves upon GAT by 1.3%, 1.6% and 2.0% on Cora, Citeseer and Pubmed, respectively. GraphSN$_{GIN}$ improves upon GIN by 3.8%, 1.7%, 1.8% and 1.6% on Cora, Citeseer, Pubmed and NELL, respectively. GraphSN$_{GraphSAGE}$ improves upon GraphSAGE by 1.3%, 1.7%, 1.1% and 2.3% on Cora, Citeseer, Pubmed and NELL, respectively.

| Method | Cora | Citeseer | Pubmed | NELL |
|---|---|---|---|---|
| GCN | $85.7 \pm 1.6$ | $73.6 \pm 1.0$ | $88.1 \pm 1.2$ | $72.2 \pm 5.6$ |
| GraphSNN$_{GCN}$ | $\mathbf{87.2 \pm 1.5}$ | $\mathbf{75.3 \pm 1.3}$ | $\mathbf{89.7 \pm 1.7}$ | $\mathbf{74.6 \pm 6.3}$ |
| GAT | $86.3 \pm 0.3$ | $74.3 \pm 0.3$ | $87.6 \pm 0.1$ | - |
| GraphSNN$_{GAT}$ | $\mathbf{87.6 \pm 0.9}$ | $\mathbf{75.9 \pm 0.8}$ | $\mathbf{89.6 \pm 0.6}$ | - |
| GIN | $82.5 \pm 0.8$ | $70.8 \pm 1.9$ | $85.0 \pm 1.5$ | $66.7 \pm 3.3$ |
| GraphSNN$_{GIN}$ | $\mathbf{86.3 \pm 0.7}$ | $\mathbf{72.5 \pm 1.5}$ | $\mathbf{86.8 \pm 1.2}$ | $\mathbf{68.3 \pm 3.7}$ |
| GraphSAGE | $86.8 \pm 1.9$ | $74.2 \pm 1.8$ | $88.3 \pm 1.1$ | $69.4 \pm 4.3$ |
| GraphSNN$_{GraphSAGE}$ | $\mathbf{88.1 \pm 1.5}$ | $\mathbf{75.9 \pm 1.3}$ | $\mathbf{89.4 \pm 2.4}$ | $\mathbf{71.7 \pm 4.5}$ |

Table 13: Classification accuracy (%) averaged over 10 random splits on node classification.

GRAPH CLASSIFICATION USING SETUP (ERRICA ET AL., 2020)

Following the data splits and experiment setup introduced in (Errica et al., 2020), we further evaluate our method. The experimental setup in (Errica et al., 2020) provides a fair performance comparison process on GNN methods. The evaluation process has two different phases: (1) model selection on the validation set, (2) model assessment on the test set. More specifically, they first split the datasets into 90 % for training and 10 % for testing. Then the entire training set is further split into 90% of training and 10% of validation. They apply the inner hold-out method to select the best model based on validation accuracy. After selecting the best model, they train the model three times on the entire training set with early stopping.

We have conducted experiments on four bioinformatics datasets (NCI1, PROTEINS, ENZYMES and D&D) and three social network datasets (COLLAB, IMDB-B and REDDIT-5k) with node features. The results of the baseline, DGCNN and GIN are taken from the paper (Errica et al., 2020). Note that the final results of DGCNN and GIN from the paper (Errica et al., 2020) are reported by computing the mean accuracy and standard deviation on the test set in these three runs, which are different from the original papers of DGCNN and GIN. Table 14 shows the results.

| Method | NCI1 | PROTEINS | ENZYMES | D&D | COLLAB | IMDB-B | REDDIT-5k |
|---|---|---|---|---|---|---|---|
| Baseline | 69.8±2.2 | $\mathbf{75.8 \pm 3.7}$ | $\mathbf{65.2 \pm 6.4}$ | $\mathbf{78.4 \pm 4.5}$ | 70.2±1.5 | 70.8±5.0 | 52.2±1.5 |
| DGCNN | 76.4±1.7 | 72.9±3.5 | 38.9±5.7 | 76.6±4.3 | 71.2±1.9 | 69.2±3.0 | 49.2±1.2 |
| GIN | 80.0±1.4 | 73.3±4.0 | 59.6±4.5 | 75.3±2.9 | 75.6±2.3 | 71.2±3.9 | 56.1±1.7 |
| GraphSNN | $\mathbf{81.6 \pm 2.8}$ | $74.5 \pm 3.5$ | $61.7 \pm 3.4$ | $77.1 \pm 3.3$ | $\mathbf{77.0 \pm 3.1}$ | $\mathbf{72.3 \pm 3.6}$ | $\mathbf{57.1 \pm 3.1}$ |

Table 14: Classification accuracy (%) averaged over 10 runs on graph classification.

GRAPH CLASSIFICATION ON OGB GRAPH DATASETS

Table 15 shows the results for the running time of the prepocessing step in our method GraphSNN for large graph datasets (averaged over 5 runs). Note that the preprocessing step can be parallellized efficiently at the node level. The CPU time shows the total preprocessing time of a dataset in which each node is preprocessed sequentially, and the CPU time per node shows the average preprocessing time per node.

OVERSMOOTHING ANALYSIS

We have also conducted further experiments to analyze the effectiveness of our method in alleviating the over-smoothing issue. We compare GIN (i.e., a spatial GNN), DFNets (Wijesinghe & Wang, 2019) (i.e., a spectral GNN), GraphSNN$_{GIN}$ and GraphSNN$_{GCN}$. For a fair comparison, we remove the dense-net architecture of DFNets and use the same hyperparameters as in the original paper. We

| Dataset | CPU time (seconds) | CPU time per node (milliseconds) |
|---|---|---|
| ogbg-molhiv | 66.97 | 0.06383 |
| ogbg-moltox21 | 79.37 | 0.54565 |
| ogbg-moltoxcast | 380.84 | 2.36417 |
| ogbg-ppa | 820.12 | 4.71235 |

Table 15: Running time of the prepocessing step for large graph datasets averaged over 5 runs.

evaluate all models over the cora dataset using the standard splits. The classification accuracy is averaged over 10 runs on node-classification.

| #Layers | GIN | GraphSNN$_{GIN}$ | DFNet | GraphSNN$_{GCN}$ |
|---|---|---|---|---|
| 1 | 73.3±1.5 | 76.1±1.6 | 80.5±0.6 | 80.1±0.8 |
| 2 | **77.6±1.3** | **79.2±1.7** | 81.9±0.5 | **83.1±1.8** |
| 3 | 75.2±1.7 | 78.5±1.3 | **82.6±0.3** | 82.0±0.8 |
| 4 | 48.6±2.1 | 77.2±2.3 | 80.7±0.6 | 80.1±0.7 |
| 5 | 40.3±1.9 | 75.9±2.1 | 75.6±0.3 | 79.1±1.2 |
| 6 | 36.1±2.3 | 73.3±1.8 | 65.3±1.3 | 76.5±1.3 |
| 7 | 27.5±2.1 | 71.9±1.5 | 60.9±1.5 | 76.3±1.3 |
| 8 | 20.3±1.8 | 69.3±2.2 | 53.6±1.3 | 75.7±1.2 |

Table 16: Oversmoothing analysis of GIN and spectral GNN (DFNet) on cora dataset.

GraphSNN can alleviate over-smoothing is because structural coefficients capture structural connectivity between a target vertex and its neighbors. Thus, a neighbor whose structural connectivity is weak would pass little message to the target vertex, whereas a neighbor whose structural connectivity is strong would pass strong message to the target vertex.

Figure 4 shows the results of GCN and GraphSNN$_{GCN}$ on the datasets Cora, Citeseer and Pubmed, in terms of classification accuracy averaged over 10 runs in the setting of standard splits.

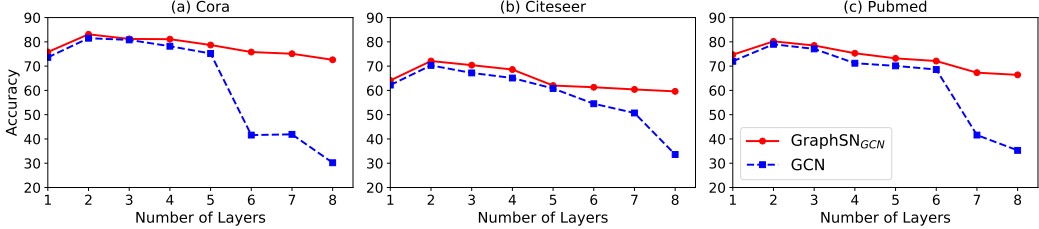

Figure 4: Oversmoothing analysis w.r.t. the model depth for node classification.

ABLATION STUDY WITH AUGMENTED NODE FEATURES

We consider an experimental evaluation setup called BL, which serves as the baseline for all experiments in this ablation study. In the setting of BL, the AGGREGATE$^I$ in GraphSNN is set to 1. Then, different variants of BL consider different local substructure counts as additional node features. This allows us to analyse what types of local substructures our proposed architecture can distinguish.

There are five variants of BL being considered in the ablation study:

(1) BL$_{SC}$: Setting AGGREGATION$^I$ of GraphSNN to 1 and keeping structural coefficients for neighbors.

(2) BL$_{NF}^{clique}$: Setting AGGREGATION$^I$ of GraphSNN to 1, removing structural coefficients for neighbors, and adding additional node features (triangle and 4-clique counts) into the original feature vectors.

| Method | GSN-v | $BL_{NF}^{clique}$ | $BL_{SC}$ | $BL_{SC+NF}^{clique}$ | GraphSNN |
|---|---|---|---|---|---|
| MUTAG | 92.20±7.5 | 90.21±2.3 | 94.06±2.4 | **95.16±2.5** | 94.70±1.9 |
| PTC-MR | 67.40±5.7 | 67.13±2.9 | 70.18±3.1 | **71.04±3.1** | 70.58±3.1 |
| PROTEINS | 74.59±5.0 | 76.42±2.6 | 78.05±2.3 | **78.66±2.1** | 78.42±2.7 |
| BZR | - | 86.82±3.1 | 90.67±3.1 | **91.98±3.2** | 91.12±3.0 |
| IMDB-B | 76.80±2.0 | 77.00±3.1 | 77.23±2.8 | **78.53±2.9** | 78.01±2.8 |

Table 17: Analysis the effects of our structural coefficients with substructure counts, i.e, triangle and 4-clique counts. Classification accuracy (%) averaged over 10 runs on graph classification.

| Method | ID-GNN | $BL_{NF}^{cycle}$ | $BL_{SC}$ | $BL_{SC+NF}^{cycle}$ | GraphSNN |
|---|---|---|---|---|---|
| MUTAG | 96.50±3.2 | 91.36±2.1 | 94.06±2.4 | **96.61±2.3** | 94.70±1.9 |
| PTC-MR | 61.90±5.4 | 67.57±3.3 | 70.18±3.1 | **71.76±3.2** | 70.58±3.1 |
| PROTEINS | 78.00±3.5 | 77.26±2.5 | 78.05±2.3 | **78.95±2.5** | 78.42±2.7 |
| BZR | 86.40±3.0 | 86.83±3.3 | 90.67±3.1 | **91.75±3.4** | 91.12±3.0 |
| IMDB-B | - | 76.36±2.6 | 77.23±2.8 | **78.58±2.4** | 78.01±2.8 |

Table 18: Analysis the effects of our structural coefficients with substructure counts, i.e, cycle counts. Classification accuracy (%) averaged over 10 runs on graph classification.

(3) $BL_{SC+NF}^{clique}$: Setting AGGREGATION$^I$ of GraphSNN to 1, keeping structural coefficients for neighbors, and adding additional node features (triangle and 4-clique counts) into the original feature vectors.

(4) $BL_{NF}^{cycle}$: Setting AGGREGATION$^I$ of GraphSNN to 1, removing structural coefficients for neighbors, and adding additional node features (cycle counts) into the original feature vectors.

(5) $BL_{SC+NF}^{cycle}$: Setting AGGREGATION$^I$ of GraphSNN to 1, keeping structural coefficients for neighbors, and adding additional node features (cycle counts) into the original feature vectors.

We compare GraphSNN with GSN-v (Bouritsas et al., 2020), $BL_{NF}^{clique}$, $BL_{SC}$, and $BL_{SC+NF}^{clique}$ to analyze how our proposed architecture relates to the models with triangle and 4 clique counts as additional node features. Similarly, we compare GraphSNN with ID-GNNs (You et al., 2021), $BL_{NF}^{cycle}$, $BL_{SC}$, and $BL_{SC+NF}^{cycle}$ to analyze how our proposed architecture relates to the models with cycle counts as additional node features. We concatenate the counts of cycles with length 1 to 4 starting and ending at the given source node with its original feature vector as in (You et al., 2021). Table 17 and Table 18 show the experimental results. As AGGREGATE$^I$ is set to 1 in the setting of BL, the performance gap between $BL_{NF}$ and $BL_{SC+NF}$ reflects the effectiveness of structural coefficients on enhancing relational inference between a target vertex and its neighbors. The performance gap between $BL_{SC}$ and GraphSNN above shows the effectiveness of AGGREGATE$^I$ in our proposed model GraphSNN. Furthermore, $BL_{SC+NF}$ consistently performs best since we incorporate both extra node features and structural coefficients into the feature aggregation. There is a small performance gap between $BL_{SC+NF}$ and GraphSNN due to augmented node features that can capture additional structural information that cannot be captured using structural coefficients.

## C. PROOFS FOR LEMMAS AND THEOREMS

**Proof for Theorem 1**

**Theorem 1.** *The following statements are true: (a) If $S_i \simeq_{subgraph} S_j$, then $S_i \simeq_{overlap} S_j$; but not vice versa; (b) If $S_i \simeq_{overlap} S_j$, then $S_i \simeq_{subtree} S_j$; but not vice versa.*

*Proof.* In the following, we prove the statements in this theorem one by one.

For Statement (a), by $S_i \simeq_{subgraph} S_j$ and Definition 1, we know that there exists a bijective mapping $g' : \tilde{\mathcal{N}}(i) \to \tilde{\mathcal{N}}(j)$ such that for the vertex $i$ and any vertex $v' \in \mathcal{N}(i)$, $i$ and $v'$ are adjacent in $S_i$ iff $j = g(i)$ and $u' = g(v')$ are adjacent in $S_j$, and $h_i = h_j$ and $h_{v'} = h_{u'}$, where $g$ is a bijective mapping between $S_i$ and $S_j$ as defined by Definition 1. Then for each pair of overlap subgraphs $S_{iv'}$

and $S_{ju'}$, we can further extend $g'$ along $g$ on $S_{iv'}$ and $S_{ju'}$. That is, $g'(v) = u$ iff $g(v) = u$. If $v$ in $S_{iv'}$, by the definition of overlap subgraph, $v$ must either be $i$ or a neighbor of $i$. Hence $u = g'(v)$ in this case must be either $j$ or a neighbor of $j$. By the definition of $g$ and the fact that $g'(v) = u$ iff $g(v) = u$, we know that for any two vertices $v_1$ and $v_2$ in $S_{iv'}$, they are adjacent in $S_{iv'}$ iff their corresponding vertices $g'(v_1)$ and $g'(v_2)$ are adjacent in $S_{ju'}$ and their corresponding feature vectors are indistinguishable, i.e, $S_{iv'} \simeq_{subgraph} S_{ju'}$ for any $v' \in \mathcal{N}(i)$ and $g(v') = u'$. Conversely, if $S_i \simeq_{overlap} S_j$, then it is possible that $S_i \not\simeq_{subgraph} S_j$ as shown by the two graphs in Figure 2(a).

For Statement (b), if $S_i \simeq_{overlap} S_j$, then to prove $S_i \simeq_{subtree} S_j$ we need to show that there exists a bijective mapping $g : \tilde{\mathcal{N}}(i) \to \tilde{\mathcal{N}}(j)$ such that $g(i) = j$ and, for any $v' \in \mathcal{N}(i)$ and $g(v') = u'$, the feature vectors of $v'$ and $u'$ are indistinguishable, i.e., $h_{v'} = h_{u'}$. By Def. 2, we can find a bijective mapping $g' : \tilde{\mathcal{N}}(i) \to \tilde{\mathcal{N}}(j)$ such that $g'(i) = j$ and, for any $v' \in \mathcal{N}(i)$ and $g'(v') = u'$, $S_{iv'}$ and $S_{ju'}$ are subgraph-isomorphic. This implies that $g'$ cannot distinguish the feature vectors of $v'$ and $u'$ for any $v' \in \mathcal{N}(i)$ and $g(v') = u'$. Similarly, the converse does not necessarily hold and one counterexample is the set of graphs as shown in Figure 2(b) which are subtree-isomorphic but not overlap-isomorphic. $\square$

## Proof for Theorem 2

**Theorem 2.** *Let $M$ be a GNN. $M$ is as powerful as 1-WL in distinguishing non-isomorphic graphs if $M$ has a sufficient number of layers and each layer can map any $S_i$ and $S_j$ in $\mathcal{S}$ into two different embeddings (i.e., $\zeta(S_i) \neq \zeta(S_j)$) if and only if $S_i \not\simeq_{subtree} S_j$.*

*Proof.* We first show that, for any two graphs $G_1$ and $G_2$, if they can be distinguished by 1-WL, then they must be distinguishable by such a GNN $M$ as well. Suppose that 1-WL takes k iterations to distinguish $G_1$ and $G_2$, i.e., 1-WL yields the same multiset of node labels on $G_1$ and $G_2$ in the iterations from 0 to $k$-1, but two different multisets of node labels on $G_1$ and $G_2$ in the k-th iteration. To derive a contradiction, we assume that a GNN $M$ that satisfies the above two conditions cannot distinguish $G_1$ and $G_2$ in the iterations from 0 to $k$. Since 1-WL can distinguish $G_1$ and $G_2$ in the k-th iteration, it means that there must exist two neighborhood subgraphs, say $S_i$ and $S_j$, which correspond to two different multisets of node labels on $G_1$ and $G_2$ at the k-th iteration. These two different multisets of node labels correspond to two different multisets of feature vectors in their neighborhoods, i.e., $\{\!\{h_v | v \in \mathcal{N}(i)\}\!\} \neq \{\!\{h_u | u \in \mathcal{N}(j)\}\!\}$. By Def. 3, we know that $S_i \not\simeq_{subtree} S_j$. Then this means that $\zeta(S_i) \neq \zeta(S_j)$, which contradicts the assumption that $M$ cannot distinguish $G_1$ and $G_2$ in the iteration $k$.

Now, we show the other direction that, for any two graphs $G_1 = (V_1, E_1)$ and $G_2 = (V_2, E_2)$, if they can be distinguished by such a GNN $M$, then they must be distinguishable by 1-WL. Similarly, suppose that at the k-th iteration, $M$ maps the neighborhood subgraphs of these two graphs into two different multisets of node embeddings, i.e., $\{\!\{\zeta(S_v) | v \in V_1\}\!\} \neq \{\!\{\zeta(S_u) | v \in V_2\}\!\}$. This is means that we can find at least two different neighborhood subgraphs $S_i$ and $S_j$ such that $\zeta(S_i) \neq \zeta(S_j)$. For such neighborhood subgraphs $S_i$ and $S_j$, we know that $S_i \not\simeq_{subtree} S_j$. Then this means that $S_i$ and $S_j$ correspond to either $h_i \neq h_j$ or $\{\!\{h_v | v \in \mathcal{N}(i)\}\!\} \neq \{\!\{h_u | u \in \mathcal{N}(j)\}\!\}$, which can be relabeled by 1-WL into two different new labels. Thus, 1-WL can also distinguish such neighborhood subgraphs, and accordingly distinguish $G_1$ and $G_2$.

The proof is completed. $\square$

## Proof for Theorem 3

**Theorem 3.** *Let $M$ be a GNN whose aggregation scheme $\Phi$ is defined by Eq. 1-Eq. 3. $M$ is strictly more expressive than 1-WL in distinguishing non-isomorphic graphs if $M$ has a sufficient number of layers and also satisfies the following conditions:*

*(1) $M$ can distinguish at least two neighborhood subgraphs $S_i$ and $S_j$ with $S_i \simeq_{subtree} S_j$, $S_i \not\simeq_{subgraph} S_j$ and $\{\!\{\tilde{A}_{iv'} | v' \in \mathcal{N}(i)\}\!\} \neq \{\!\{\tilde{A}_{ju'} | u' \in \mathcal{N}(j)\}\!\}$;*

*(2) $\Phi\left(h_v^{(t)}, \{\!\{h_u^{(t)} | u \in \mathcal{N}(v)\}\!\}, \{\!\{(\tilde{A}_{vu}, h_u^{(t)}) | u \in \mathcal{N}(v)\}\!\}\right)$ is injective.*

*Proof.* We prove this theorem in two steps. First, we prove that a GNN $M$ satisfying the above conditions can distinguish any two graphs that are distinguishable by 1-WL by contradiction. Assume that there exist two graphs $G_1$ and $G_2$ which can be distinguished by 1-WL but cannot be distinguished by $M$. Further, suppose that 1-WL cannot distinguish these two graphs in the iterations from 0 to $k$-1, but can distinguish them in the k-th iteration. Then, there must exist two neighborhood subgraphs $S_i$ and $S_j$ whose neighboring nodes correspond to two different multisets of node labels at the k-th iteration, i.e., $\{\!\{h_v^{(k)}|v \in \mathcal{N}(i)\}\!\} \neq \{\!\{h_u^{(k)}|u \in \mathcal{N}(j)\}\!\}$. By the above condition (2), we know that $\Phi$ is injective. Thus, for $S_i$ and $S_j$, $\Phi$ would yield two different feature vectors at the k-th iteration. This means that $M$ can also distinguish $G_1$ and $G_2$, which contradicts the assumption. Our proof in the first step is done. For the second step, we can prove that there exist at least two graphs that can be distinguished by $M$ but cannot be distinguished by 1-WL. Figure 1 presents two of such graphs. □

**Proof for Theorem 4**

We consider that, for each vertex in a graph, its node features are from a countable set; similarly, for each pair of adjacent vertices in a graph, its structural coefficient is also from a countable set. Assume that $\mathcal{H}$, $\mathcal{A}$ and $\mathcal{W}$ are countable sets where $\mathcal{H}$ is a node feature space, $\mathcal{A}$ is a structural coefficient space, and $\mathcal{W} = \{A_{ij}h_i|A_{ij} \in \mathcal{A}, h_i \in \mathcal{H}\}$. Let $H$ and $W$ be two multisets containing elements from $\mathcal{H}$ and $\mathcal{W}$, respectively, and $|H| = |W|$.

**Lemma 1.** *There exists a function $f$ s.t. $\pi(H, W) = \sum_{h \in H, w \in W} f(h, w)$ is unique for any distinct pair of multisets $(H, W)$.*

*Proof.* Since $\mathcal{H}$ and $\mathcal{W}$ are countable, there must exist two functions $\psi_1 : \mathcal{H} \to \mathbb{N}_{odd}$ mapping $h \in \mathcal{H}$ to odd natural numbers and $\psi_2 : \mathcal{W} \to \mathbb{N}_{even}$ mapping $w \in \mathcal{W}$ to even natural numbers. Further, for any pair of multisets $(H, W)$, since the cardinality of $H$ and $W$ is bounded, there must exist a number $N \in \mathbb{N}$ such that $|H| < N$ and $|W| < N$. Thus, we can find a prime number $P > 2N$. Then we have a mapping $f$ as $f(h, w) = P^{-\psi_1(h)} + P^{-\psi_2(w)}$ such that $\sum_{h \in H, w \in W} f(h, w)$ is unique for each distinct pair of $(H, W)$. □

**Lemma 2.** *There exists a function $f$ s.t. $\pi'(h_v, H, W) = \gamma f(h_v, |H|h_v) + \sum_{h \in H, w \in W} f(h, w)$ is unique for any distinct $(h_v, H, W)$, where $h_v \in \mathcal{H}$, $|H|h_v \in \mathcal{W}$, and $\gamma$ can be an irrational number.*

*Proof.* As $h_v \in \mathcal{H}$ and $|H|h_v \in \mathcal{W}$, we may have $f(h_v, |H|h_v) = P^{-\psi_1(h_v)} + P^{-\psi_1(|H|h_v)}$ where $\psi_1 : \mathcal{H} \to \mathbb{N}_{odd}$ and $\psi_2 : \mathcal{W} \to \mathbb{N}_{even}$ as defined in the proof for Lemma 1. Let $(h_{v1}, H_1, W_1)$ and $(h_{v2}, H_2, W_2)$ be two different tuples. Then, there are two cases:

(1) When $h_{v1} = h_{v2}$ but $(H_1, W_1) \neq (H_2, W_2)$, by Lemma 1, we know that $\sum_{h \in H_1, w \in W_1} f(h, w) \neq \sum_{h \in H_2, w \in W_2} f(h, w)$. Thus, $\pi'(h_{v1}, H_1, W_1) \neq \pi'(h_{v2}, H_2, W_2)$.

(2) When $h_{v1} \neq h_{v2}$, we prove $\pi'(h_{v1}, H_1, W_1) \neq \pi'(h_{v2}, H_2, W_2)$ by contradiction. Assume that $\pi'(h_{v1}, H_1, W_1) = \pi'(h_{v2}, H_2, W_2)$. Then, we have:

$$\gamma f(h_{v1}, |H_1|h_{v1}) + \sum_{h \in H_1, w \in W_1} f(h, w) = \gamma f(h_{v2}, |H_2|h_{v2}) + \sum_{h \in H_2, w \in W_2} f(h, w).$$

This gives us the following equation:

$$\gamma \Big( f(h_{v1}, |H_1|h_{v1}) - f(h_{v2}, |H_2|h_{v2}) \Big) = \Big( \sum_{h \in H_2, w \in W_2} f(h, w) \Big) - \Big( \sum_{h \in H_1, w \in W_1} f(h, w) \Big).$$

When $\gamma$ is an irrational number, L.H.S. of the above equation is irrational but R.H.S. is rational. There is a contradiction. Thus, $\pi'(h_{v1}, H_1, W_1) \neq \pi'(h_{v2}, H_2, W_2)$.

□

Based on Lemma 1 and Lemma 2, we can prove the following theorem.

**Theorem 4.** *GraphSNN is more expressive than 1-WL in testing non-isomorphic graphs.*

*Proof.* We prove this theorem by showing that GraphSNN is a GNN satisfying the conditions stated in Theorem 3. For the first condition, consider the two graphs shown in Figure 1. GraphSNN can distinguish these two neighborhood subgraphs $S_i$ and $S_j$ with $\{\!\!\{\tilde{A}_{iv'}|v' \in \mathcal{N}(i)\}\!\!\} \neq \{\!\!\{\tilde{A}_{ju'}|u' \in \mathcal{N}(j)\}\!\!\}$. For the second condition, by Lemmas 1 and 2 as well as the fact that MLP as a universal approximator (Xu et al., 2019) can be used to model and learn the functions $f$ and $g$, we know that GraphSNN also satisfies this condition. □

