# OpenReview forum: "A New Perspective on "How Graph Neural Networks Go Beyond Weisfeiler-Lehman?""
_ICLR.cc/2022/Conference — ICLR 2022 Oral_

### Official Review · Reviewer_Z8WJ · 2021-11-02

**Correctness:** 4
**Technical Novelty And Significance:** 3
**Empirical Novelty And Significance:** Not applicable
**Recommendation:** 8
**Confidence:** 3

**Main Review:**

1. **Originality**

The paper belongs to a body of recent works which aim to break the expressive power limit of MPNNs in a sub-quadratic memory complexity in the number of nodes. As far as I’m aware, the aggregation scheme suggested in the paper is novel, providing a constructive way to design more powerful architectures than the 1-WL test.

3. **Experimental evaluation**

   ***Strengths*** - the paper demonstrates a boost in performance on node and graph classification.

   ***Weaknesses*** -

   1. Expressive power experiments  - also an experimental evaluation showing what types of structures the proposed architecture can distinguish would be interesting to see (e.g., cycles, d-regular graphs).
   2. Over-smoothing - the empirical results are not explained. Do the authors have a conjecture as to why their proposed method circumvents over smoothing?

   ***Suggestions***

   - Further discussion on generalization to unseen graphs (With larger / smaller average neighborhoods)? Since the aggregation coefficients suggested are normalized (edges/vertices) it might be that the suggested model is more robust to size differences in graphs. It would be interesting to see how it performs. For example, a simple experiment like the ones performed in [1].


- **Clarity** - paper is well written; claims are well supported by theoretical proofs and guarantees.


[1] Yehudai, G., Fetaya, E., Meirom, E., Chechik, G., & Maron, H. (2020). On Size Generalization in Graph Neural Networks. ArXiv, abs/2010.08853.





**Summary Of The Paper:**

The paper introduces a general framework for designing message-passing neural networks (MPNNs) stronger than the 1-WL distinguishing power via structural information injection to the aggregation scheme in the message passing framework. Unlike common MPNNs which perform neighborhood aggregation based on a pre-defined fixed function (which does not depend on the neighborhood subgraph structure) or based on a learned function of the neighborhood node features, the paper suggests a weighting function in the aggregation step of MPNNs which depends on the structure of the neighborhood subgraph. The authors show that under reasonable conditions the resulting architecture has a superior expressive power than the 1-WL test. This also implies superiority in expressive power over common MPNNs while maintaining the same memory and time complexity. The paper introduces a hierarchy of 3 levels of neighborhood isomorphisms: subtree, overlap, and subgraph; which facilitate the proofs and theoretical guarantees. Furthermore, as a by-product, the authors show improved robustness to the notorious over smoothing problem in MPNNs.

**Summary Of The Review:**

A well written paper with strong theoretical exposition and results.


---
**Post Rebuttal**: I am satisfied with the authors response and keep my original score.

---

> ### Author Response · Authors · 2021-11-17
> **Author Responses to Reviewer Z8WJ**
>
> Thanks for the comments.
>
> Q1. Expressive power experiments - also an experimental evaluation showing what types of structures the proposed architecture can distinguish would be interesting to see (e.g., cycles, d-regular graphs).
>
> We have added the experimental results and discussion into Appendix B in the revised version.
>
> We consider an experimental evaluation setup called BL, which serves as the baseline for all experiments in this ablation study. In the setting of BL, the AGGREGATE$^I$ in GraphSNN is set to 1. Then, different variants of BL consider different local substructure counts as additional node features. This allows us to analyse what types of local substructures our proposed architecture can distinguish.
>
> There are five variants of BL being considered in the ablation study:
>
> > BL$_{SC}$: Setting AGGREGATION$^I$ of GraphSNN to 1 and keeping structural coefficients for neighbors.
>
> > BL$^{clique}_{NF}$: Setting AGGREGATION$^I$ of GraphSNN to 1, removing structural coefficients for neighbors, and adding additional node features (triangle and 4-clique counts) into the original feature vectors.
>
> > BL$^{clique}_{SC+NF}$: Setting AGGREGATION$^I$ of GraphSNN to 1, keeping structural coefficients for neighbors, and adding additional node features (triangle and 4-clique counts) into the original feature vectors.
>
> > BL$^{cycle}_{NF}$: Setting AGGREGATION$^I$ of GraphSNN to 1, removing structural coefficients for neighbors, and adding additional node features (cycle counts) into the original feature vectors.
>
> > BL$^{cycle}_{SC+NF}$: Setting AGGREGATION$^I$ of GraphSNN to 1, keeping structural coefficients for neighbors, and adding additional node features (cycle counts) into the original feature vectors.
>
> We compare GraphSNN with GSN-v [1], BL$^{clique}_{NF}$, BL$\_{SC}$ , and BL$^{clique}\_{SC+NF}$ to analyze how our proposed architecture relates to the models with triangle and 4 clique counts as additional node features.
>
> Similarly, we compare GraphSNN with ID-GNNs [2], BL$^{cycle}\_{NF}$, BL$\_{SC}$, and BL$^{cycle}\_{SC+NF}$ to analyze how our proposed architecture relates to the models with cycle counts as additional node features. We concatenate the counts of cycles with length 1 to 4 starting and ending at the given source node with its original feature vector as in [2].
>
> Table 1 and Table 2 show the experimental results. As AGGREGATE$^I$ is set to 1 in the setting of BL, the performance gap between BL$\_{NF}$ and BL$\_{SC+NF}$ reflects the effectiveness of structural coefficients on enhancing relational inference between a target vertex and its neighbors. The performance gap between BL$\_{SC}$ and GraphSNN below shows the effectiveness of AGGREGATE$^I$ in our proposed model GraphSNN. Furthermore, BL$\_{SC+NF}$ consistently performs best since we incorporate both extra node features and structural coefficients into the feature aggregation. There is a small performance gap between BL$\_{SC+NF}$ and GraphSNN due to augmented node features that can capture additional structural information that cannot be captured using structural coefficients.
>
> |Dataset    |GSN-v [1]       | BL$^{clique}_{NF}$          | BL$_{SC}$            | BL$^{clique}_{SC+NF}$        |GraphSNN        |
> |:---       |:----:         |:----:         |:----:           |:----:           |:----:         |
> |MUTAG      |92.20$\pm$7.5   |90.21$\pm$2.3  |94.06$\pm$2.4    |95.16$\pm$2.5    |94.70$\pm$1.9|
> |PTC-MR     |67.40$\pm$5.7   |67.13$\pm$2.9 |70.18$\pm$3.1    |71.04$\pm$3.1    |70.58$\pm$3.1|
> |PROTEINS   |74.59$\pm$5.0   |76.42$\pm$2.6 |78.05$\pm$2.3    |78.66$\pm$2.1   |78.42$\pm$2.7|
> |BZR        |    -          |86.82$\pm$3.1  |90.67$\pm$3.1    |91.98$\pm$3.2    |91.12$\pm$3.0|
> |IMDB-B     |76.80$\pm$2.0   |77.00$\pm$3.1  |77.23$\pm$2.8    |78.53$\pm$2.9    |78.01$\pm$2.8|
>
>
> |Dataset    |ID-GNN [2]       | BL$^{cycle}_{NF}$          | BL$_{SC}$            | BL$^{cycle}_{SC+NF}$         |GraphSNN        |
> |:---       |:----:         |:----:         |:----:           |:----:           |:----:         |
> |MUTAG      | 96.50$\pm$3.2 |91.36$\pm$2.1  |94.06$\pm$2.4    |96.61$\pm$2.3    |94.70$\pm$1.9|
> |PTC-MR     | 61.90$\pm$5.4   |67.57$\pm$3.3 |70.18$\pm$3.1    |71.76$\pm$3.2    |70.58$\pm$3.1|
> |PROTEINS   | 78.00$\pm$3.5   |77.26$\pm$2.5 |78.05$\pm$2.3    |78.95$\pm$2.5   |78.42$\pm$2.7|
> |BZR        | 86.40$\pm$3.0       |86.83$\pm$3.3  |90.67$\pm$3.1    |91.75$\pm$3.4    |91.12$\pm$3.0|
> |IMDB-B     |       -       |76.36$\pm$2.6  |77.23$\pm$2.8    |78.58$\pm$2.4    |78.01$\pm$2.8|
>
> [1] Improving graph neural network expressivity via subgraph isomorphism counting. Giorgos et al., arxiv.
>
> [2] Identity-Aware Graph Neural Networks. Jiaxuan You et al., AAAI 2021.

---

> > ### Author Response · Authors · 2021-11-17
> > **Author Responses to Reviewer Z8WJ**
> >
> > Q2. Over-smoothing - the empirical results are not explained. Do the authors have a conjecture as to why their proposed method circumvents over smoothing?
> >
> > The reason why our method GraphSNN can alleviate over-smoothing is because structural coefficients capture structural connectivity between a target vertex and its neighbors. Thus, a neighbor whose structural connectivity is weak would pass little messages to the target vertex, whereas a neighbor whose structural connectivity is strong would pass a strong message to the target vertex.
> >
> > Thanks for the suggestions on generalization to unseen graphs. This is an interesting research direction. We will explore it in our future work.

---

> > > ### Comment · Reviewer_Z8WJ · 2021-11-29
> > > **Satisfied with the answers**
> > >
> > > I thank the authors for their response.
> > > I am satisfied with the answers and think the scores the paper received are well deserved.
> > > I keep my original score.

---

### Official Review · Reviewer_HLe3 · 2021-11-02

**Correctness:** 4
**Technical Novelty And Significance:** 3
**Empirical Novelty And Significance:** 3
**Recommendation:** 8
**Confidence:** 4

**Main Review:**

* Pros:

1. Overlap isomorphism is a nice point to find a balance between expressive power and computational efficiency. Based on that, it is persuasive to encorporate the three properties into structural coefficients.
2. The setting of $\lambda$ is very interesting to me. It brings the coefficients some flexibility so that different $\lambda$'s may capture inherent key information in different learning tasks, as shown in experiments.
3. GraphSNN shows impressive experiment results. On node classification, it improves the counterparts of traditional GNNs by considerable boost, but is also very easy to use. On graph classification, it outperforms baselines by an intriguingly large margin. The performance drops slowly when it stacks more layers.

* Concerns:

1. In the beginning of page 2, the sentence ``compared with the methods of augmenting node identifiers ...'' needs more explanation. If I understand correctly, the latter claim is from the ablation study of $\lambda$ in Section 5.3. But I do not see its relationship with node id/random feature models.
2. According to the definition of $\tilde{A}_{vu}$ with normalization above (5), can the first summation over $u\in \mathcal{N}(v)$ be simplified as 1?
3. Since structural coefficients emphasize strongly connected neighborhood, I am wondering whether it would hurt the performance when there is a task requiring long-range information [1] and the path is, to some extent, adversarially going through a path with weaker connectivity.
4. With respect to oversmoothing on node classification, I am wondering whether the graph operator (can be derived from (5)) with different $\lambda, \gamma$ has dominating subspace [3] that aligns well with the labels. A possible experiment is to send features through leading eigenvectors and then do pure MLP, following [2].
5. Expressive GNNs typically show better performance on graph regression tasks than graph classification. So it would be better to show comparision with expressive baselines on regression datasets such as QM9 and ZINC, instead of Table 4, if time permits.

Reference:

[1] On the bottleneck of graph neural networks and its practical implications. U Alon, E Yahav.

[2] Revisiting Graph Neural Networks: All We Have is Low-Pass Filters. NT Hoang, T Maehara.

[3] Graph Neural Networks Exponentially Lose Expressive Power for Node Classification. Kenta Oono, Taiji Suzuki.

**Summary Of The Paper:**

To overcome expressive weakness of message passing neural networks and computational efficiency of expressive GNNs, this paper proposes a new perspective by introducing overlap subgraphs and overlap isomorphism which is between subgraph isomorphism and subtree isomorphism. Based on that, it carefully design GraphSNN with local structural coefficients to control message passing to obtain more expressive power than 1-WL GNNs. Experiments are conducted on node-level and graph-level classification tasks with an abalation study on a key hyperparameter.

**Summary Of The Review:**

I would like to recommend to accept this paper, for its expressive, efficient and easy-to-use GNN component with intriguing performance.

---

> ### Author Response · Authors · 2021-11-17
> **Author Responses to Reviewer HLe3**
>
> Thanks for the suggestions. They're very constructive.
>
> Q1. In the beginning of page 2, the sentence ``compared with the methods of augmenting node identifiers ...'' needs more explanation. If I understand correctly, the latter claim is from the ablation study of  in Section 5.3. But I do not see its relationship with node id/random feature models.
>
> ID-GNN [1] counts the number of cycles at each level of the computation graph, augments it with the root node's feature vector, and then simply applies the homogeneous message passing scheme. In the work of Sato et al. [2], i.e., the earliest work that has considered the effects of random features on the expressive power of GNNs, it has shown that the expressive power of GNNs can be improved by adding random features to each node. For instance, if each node is assigned a random feature, GNNs can determine the existence of a cycle of length m by checking whether there exists the same value as the root node at depth m. Thus, random features are considered as being able to identify nodes and accordingly identify cycles.
>
> In our work, we don't explicitly augment these kinds of features. Furthermore, structural coefficients in our work can flexibly quantify various local structures in an overlap graph, and can also capture structural properties for different graph learning tasks by adjusting the value of $\lambda$. We have added an explanation in the revised version.
>
> [1] Identity-Aware Graph Neural Networks, Jiaxuan You et al., AAAI 2021.
>
> [2] Random Features Strengthen Graph Neural Networks, Sato et al.
>
> Q2. According to the definition of $\tilde{A}_{vu}$ with normalization above (5), can the first summation over $u \in \mathcal{N}(v)$ be simplified as 1?
>
> The first summation of Eq. 5 can be simplified as $\Big( \sum_{u \in \mathcal{N}(v)}  \tilde{A}\_{vu} + 1\Big)=1+|N_v|$, where $|N_v|$ represents the number of neighbors of vertex $v$. The reason why we add 1 for each neighborhood structural coefficient $\tilde{A}_{vu}$ in this term is to ensure the injectivity of the feature aggregation in the presence of structural coefficients.
>
> Q3. Since structural coefficients emphasize strongly connected neighborhood, I am wondering whether it would hurt the performance when there is a task requiring long-range information [1] and the path is, to some extent, adversarially going through a path with weaker connectivity.
>
> We need multiple GNN layers for long-range tasks. A node's receptive field grows exponentially with the number of layers in some of the GNN models such as GCN and GIN. Therefore, GCN and GIN suffers from the over-squashing problem since every node of them performs the direct summation of its neighbors. However, our method GraphSNN can alleviate this issue, because structural coefficients capture structural connectivity between a target vertex and its neighbors. Thus, a neighbor whose structural connectivity is weak would pass little messages to the target vertex, whereas a neighbor whose structural connectivity is strong would pass a strong message to the target vertex. Generally, studying tasks that require long-range information which is adversarially going through a path with weaker connectivity is an interesting research direction. We will explore it further.
>
> Q4. With respect to oversmoothing on node classification, I am wondering whether the graph operator (can be derived from (5)) with different $\lambda$ and $\gamma$ has dominating subspace [3] that aligns well with the labels. A possible experiment is to send features through leading eigenvectors and then do pure MLP, following [2].
>
> Our model does not behave like a spectral GNN, and has a different way to leverage structural information. Concretely, spectral GNNs leverage the spectrum of Laplacian through low-pass/high pass filters which cannot distinguish how a target vertex is structurally connected to each of its neighbors. In contrast, our model can capture such structural connectivity information using structural coefficients defined on overlap subgraphs. We observe that the structural connectivity information is important for alleviating over-smoothing. We have conducted an experiment to compare the overmoothing of spectral GNNs by DFNets with our proposed method GraphSNN (please refer over-smoothing analysis section in Appendix B for the experimental results of DFNet and GraphSNN). The results show that DFNet suffers more with over-smoothing when increasing the layers.
>
> Q5. Expressive GNNs typically show better performance on graph regression tasks than graph classification. So it would be better to show comparision with expressive baselines on regression datasets such as QM9 and ZINC, instead of Table 4, if time permits.
>
> Thanks for your suggestion. We may need some time to implement and evaluate it for graph regression problem. We are planning for future work to evaluate GraphSNN on regression datasets such as QM9 and ZINC.

---

> > ### Comment · Reviewer_HLe3 · 2021-11-29
> > **Reviewer response**
> >
> > I would like to keep my score.

---

### Official Review · Reviewer_FAad · 2021-11-05

**Correctness:** 4
**Technical Novelty And Significance:** 3
**Empirical Novelty And Significance:** 3
**Recommendation:** 8
**Confidence:** 4

**Main Review:**

This paper proposes an efficient method for message passing that can incorporate structural information (that of neighborhood subgraphs) that is provably more expressive than 1-WL. As compared to three strands of provably powerful (more than 1 WL) GNNs, the method has limited additional computational overhead, and can also show encouraging results on the oft-quoted "over smoothing" problem. This is borne out in extensive experimental results and a satisfactory ablation.

In the recent literature on GNNs, there have emerged three directions of research that aim to construct procedures more expressive than 1 WL:
1. Higher-order WL-based methods, which often can get prohibitively expensive.
2. Using features generated from substructures (most of the papers taking this approach claim this information is available through domain knowledge, but often just use triangles and the likes).
3. Augmenting node features with identifying information to improve expressive power.

The central idea of the paper is grounded in the common observation that treating the neighborhood as a multiset of features ignores rich topological information, limiting the expressivity of message passing procedures that use such a representation. If the neighborhood is represented as a neighborhood subgraph, then 1-WL is only as powerful as distinguishing neighborhood subgraphs in terms of subtree structures.  The question then becomes if structural information can be incorporated in a way that can go beyond neighborhood subtree isomorphism. Towards this end, the authors show that there exists a class of isomorphic graphs that lie in between neighborhood subgraph isomorphism and neighborhood subtree isomorphism, which they call overlap subgraph isomorphism. It is shown that by incorporating structural information that can solve overlap subgraph isomorphism, one gets a message-passing network that is more expressive than WL.

To be more specific, section 2 provides the hierarchy of local isomorphism mentioned above. Theorem 1 states that subgraph isomorphism implies overlap subgraph isomorphism but not vice-versa, and that overlap subgraph isomorphism implies subtree-isomorphism but not vice-versa. 1-WL can only distinguish those graphs that can solve for sub-tree isomorphism at each layer. In order to go beyond 1-WL, the authors focus on overlap-subgraph isomorphism and define a set of structural coefficients for each vertex based on overlap subgraphs. These coefficients depend on reasonable notions of closeness, density, and invariance. The exact form of such coefficients is shown in section 4, and it is also shown that incorporating such information can give a method strictly more powerful than 1 WL. Since it just admits the same message passing paradigm, the computational overhead is limited.

Extensive experiments on node classification (Cora, CIteseer, Pubmed, NELL, ogbn-arxiv), graph classification for the commonly used small graph datasets, and large graphs show across-the-board improvement compared to the competition (including the higher-order methods). The ablation shows the importance of the structural coefficients. Further, another set of experiments show that the proposed method is able to avoid the so-called oversmoothing problem (however, the results are presented without comment -- it would be beneficial to share some intuition on why this is the case).

In summary, I think the paper makes a solid contribution. It proposes a well-motivated method and validates it by extensive experimentation that leaves little doubt on its efficacy.

Minor comments:
- The paper will benefit from sharpening the writing in the abstract and intro -- it feels a little bit wayward and has some convoluted sentences.
- The paper title, and most of the paper itself, uses the spelling "Lehman." The actual spelling is Leman. Somewhere after the intro, when the original WL paper is cited (and in some following sentences), the correct spelling is used. I would suggest either using the correct spelling throughout, or sticking to Lehman throughout (since it is widely used, and Leman himself did not mind it https://www.iti.zcu.cz/wl2018/pdf/leman.pdf)
- I don't believe that the citation of "Provably Powerful Graph Networks' of Maron et al. is accurate. It is cited as a powerful (3 WL) method that is expensive, however, I think it is an efficient method. The authors might want to verify this claim.
- Line 4 from the bottom of page 1: typo "This solution enables GNNs to provably more expressive": -> "This solution enables GNNs to provably be more expressive"
- Line 2 from the bottom of page 1: "which require high computational overheads and are impractical" -> "which require high computational overhead and are impractical"
- Line 1 page 2: "(2), our method does not require any domain" I am not sure it is accurate to say that these methods require domain knowledge. They can certainly benefit from it, and this claim is made in those papers, but they just stick to simple sub-structures such as triangles that are easy to treat. This repeats on page 3.
- Line 3 of the second paragraph, page 2: "capacity of a model" --> "capacity of the model"

Disclaimer: I have not verified the proofs for correctness. However, based on the statements of the theorems and the general idea, I can buy them to be true and base my review on that assumption (since I expect these statements to be true).

**Summary Of The Paper:**

This paper proposes an efficient method for message passing that can incorporate structural information (that of neighborhood subgraphs) that is provably stronger than 1-WL. As compared to three strands of provably powerful (more than 1 WL) GNNs, the method has limited additional computational overhead, and can also show encouraging results on the oft-quoted "over smoothing problem". The benefit of better expressivity coupled with the simplicity of the method is borne out in extensive experimental results and a satisfactory ablation.

**Summary Of The Review:**

Well motivated method that is provably more powerful than 1-WL, which incorporates structural information in a principled manner using the notion of overlapping-subgraph ismorphism. Extensive experimentation shows strong performance compared to the competition (including the higher-order WL procedures).

---

> ### Author Response · Authors · 2021-11-17
> **Author Responses to Reviewer FAad**
>
> Thanks for the comments. The reason why our method GraphSNN can alleviate over-smoothing is because structural coefficients capture structural connectivity between a target vertex and its neighbors. Thus, a neighbor whose structural connectivity is weak would pass little messages to the target vertex, whereas a neighbor whose structural connectivity is strong would pass a strong message to the target vertex. We have added the explanation into Section 5.4 in the revised version.
>
> For the minor comments, below are our responses:
>
> Q1. We have revised the writing in the abstract and introduction.
>
> Q2. Thanks for pointing out the spelling inconsistencies of "Lehman" and "Leman". We have checked the paperand the revised version sticks to Lehman throughout.
>
> Q3. I don't believe that the citation of "Provably Powerful Graph Networks' of Maron et al. is accurate. It is cited as a powerful (3 WL) method that is expensive, however, I think it is an efficient method. The authors might want to verify this claim.
>
> We have revised the claim to make it more accurate.
>
> Q4. For the typos on Pages 1 and 2, thanks for pointing them out. We have fixed them in the revised version.
>
> Q5. If no domain knowledge is required, then various simple sub-structures would need to be considered, for example, triangles, 4-cliques, 5-cliques, ..., paths of length 2, length 3, ..., cycles of length 4, length 5, ..., 3-star, 4-star, ... or simply subgraphs with n vertices. This would lead to inefficiency and affect effectiveness as well. Thus, domain expert knowledge is often needed by these methods so as to choose only certain substructures of interest. Below are some examples:
>
> LRP [1] counts the number of times a given pattern such as 3-star, triangle, tailed triangle, chordal cycle and attributed triangle appears as a subgraph or induced subgraph in the original graph.
>
> GSN [2] incorporates handcrafted topological features such as the presence of cliques or cycles, which requires expert knowledge on what features are relevant for a given task [3].
>
> [1] Can Graph Neural Networks Count Substructures? Chen et al., NeurIPS 2020.
>
> [2] Improving Graph Neural Network Expressivity via Subgraph
> Isomorphism Counting, Bouritsas et al., arXiv.
>
> [3] Building powerful and equivariant graph neural networks with structural message-passing, Vignac et al., NeurIPS 2020.

---

### Official Review · Reviewer_csQY · 2021-11-07

**Correctness:** 4
**Technical Novelty And Significance:** 3
**Empirical Novelty And Significance:** 3
**Recommendation:** 8
**Confidence:** 3

**Main Review:**

The contribution of the work is definitely relevant and significant. The framework is solid, well-justified, and supported by proofs. The experimental results are quite convincing as they confirm the good performance of the method in classical datasets.

I would however encourage the authors to work a bit more on the presentation of the paper. While the paper is generally well-written, there are parts that are difficult to follow for someone who is familiar with GNNs, but not working exactly on aspects of expressivity. For example, the notions of subgraph-isomorphic, overall-isomorphic, and subtree-isomorphic are very technical, and hard to follow. I would appreciate it if the authors could make them more clear, by for example, explaining better Fig. 2.

Similarly, the proposed choice of \omega, in Eq. 4, should be better explained/motivated.

Some additional minor comments:
   - '0.05 level of significance': how do you define it? Please elaborate.
   - Any intuition on what the performance on the MUTAG dataset is not that great?
   - Please double check the References to make them complete and consistent (e.g., Waiss Azizian et al..., Cosro....'33'?,2020)



**Summary Of The Paper:**

This paper deals with the very challenging and important problem of designing Graph Neural Networks (GNNs) that are more expressive. The authors propose a new GNN framework, that injects structural information into a message-passing aggregation scheme. The proposed architecture (GraphSNN) is shown to be more expressive than the 1-WL in distinguishing graph structure. Finally, the framework is validated in state-of-the-art node classification and graph classification benchmarks.

**Summary Of The Review:**

This is a good paper, with some aspects of the presentation that should be improved.

---

> ### Author Response · Authors · 2021-11-17
> **Author Responses to Reviewer csQY**
>
> Thanks for the suggestions on improving the presentation. We have revised Fig 2 to make it more comprehensive, and also added some explanation for $\omega$ in Eq. 4.
>
> For the minor comments, below are our responses:
>
> Q1. '0.05 level of significance': how do you define it? Please elaborate.
>
> In our work, '0.05 level of significance' corresponds to 95\% confidence level, which is commonly used in the literature (e.g. [1,2,3]) to ensure the statistical robustness of experimental results.
>
> [1] Adaptive Universal Generalized PageRank Graph Neural Network, Chien et al., ICLR 2020.
>
> [2] Learning from the Past: Continual Meta-Learning with Bayesian Graph Neural Networks, Luo et al., AAAI 2020.
>
> [3] Predict then Propagate: Graph Neural Networks meet Personalized PageRank, Klicpera et al., ICLR 2019.
>
>
> Q2. Any intuition on what the performance on the MUTAG dataset is not that great?
>
> MUTAG dataset is the smallest dataset that we use in our experiments on graph classification tasks. We don't have concrete ideas but some intuition may include: (1) small number of available graphs in total, i.e., only 188 graphs with 18 nodes and 20 edges on average for each graph; (2) small number of graphs for validation, i.e., only 19 graphs are used for validation in a single fold; (3) the classes of graphs are unbalanced, i.e., the majority class is twice as the minority class.
>
> Q3. Please double check the References to make them complete and consistent (e.g., Waiss Azizian et al..., Cosro....'33'?,2020)
>
> We have checked the References and fixed it in the revised version.

---

> > ### Comment · Reviewer_csQY · 2021-11-29
> > **After rebuttal**
> >
> > I thank the authors for addressing my comments. I appreciate also their response to the other reviewers' comments. I still believe it is a good paper, and I keep my score.

---

### Public Comment · ~Ralph_Abboud1 · 2021-11-12
**Related Work**

Interesting paper! We would like to point the authors to our related work on random node initialization: https://arxiv.org/pdf/2010.01179.pdf

---

> ### Author Response · Authors · 2021-11-21
> **Author Response**
>
> Thanks for pointing out this paper. We will read it and have a check.

---

### Public Comment · ~Hanqing_Zeng1 · 2021-11-19
**Related Work**

Thanks for this interesting new perspective!

I would like to point out our related work in NeurIPS 2021: "Decoupling the depth and scope of Graph Neural Networks" (https://papers.nips.cc/paper/2021/file/a378383b89e6719e15cd1aa45478627c-Paper.pdf). Our shaDow-GNN model also provably exceeds 1-WL by utilizing local subgraph structure, while being highly efficient and scalable.

---

> ### Author Response · Authors · 2021-11-21
> **Author Response**
>
> Thanks for pointing out this paper. We will read it and have a check.

---

### Public Comment · ~Muhan_Zhang1 · 2021-11-20
**Related Work**

Thanks for the interesting work! Another related work is "Nested Graph Neural Networks" from NeurIPS 2021 (https://arxiv.org/pdf/2110.13197.pdf). It also uses subgraphs beyond subtrees to increase the expressive power of GNNs, and provably discriminates almost all regular graphs that 1-WL always fails.

---

> ### Author Response · Authors · 2021-11-21
> **Author Response**
>
> Thanks for pointing out this paper. We will read it and have a check.

---

### Public Comment · ~Lingxiao_Zhao1 · 2022-01-28
**Problems about the results on TUDatasets**

Congratulation to the great work!

I would like to point an issue of the results on TUDatasets.
I have checked the code that the reported accuracy is based on the average of 10 maximum validation accuracy on each fold (this value is higher than it should be).
However all baselines are calculating the validation accuracy by  1) first averaging 10 validation accuracy curves, 2) picking up the epoch based on the best accuracy on the **averaged validation curve**, 3) report the mean accuracy and std based on the selected epoch. I noticed this issue because I also made the mistake before in our submission but corrected it later.

Although we all know that there are lots of problems of these small TUDatasets and the evaluation procedure we followed is also not the "best", I still want to mention this issue so that the author can revise the result to keep a fair comparison. This is also important for future researchers to cite the correct result of the paper. Nevertheless this is still a great work!

---

> ### Public Comment · ~Asiri_Wijesinghe1 · 2022-02-03
> **Author Response**
>
> Hi Lingxiao,
>
> Thank you very much for pointing this out to us!
>
> We will check the reporting settings of all the baselines and revise the results to ensure a fair and consistent comparison in our final paper.

---

### Public Comment · ~Jiacheng_You1 · 2022-02-06
**add 1 design is not discussed.**

In Section 4,
"Note that, to ensure the injectivity in the feature aggregation in the presence of structural coefficients, we add 1 into the first and second terms in Eq. 5. This design is critical for guaranteeing the expressiveness of GraphSNN beyond 1-WL, as will be discussed in the proofs of the lemmas and Theorem 4 later."
However, this design is not discussed in appendix C.

BTW, in the proof of Theorem 4, "For the second condition, by Lemmas 1 and 2 as well as the fact that MLP as a universal
approximator (Xu et al., 2019) can be used to model and learn the functions f and g, we know that
GraphSNN also satisfies this condition."
However, Lemmas 1 and 2 only use f, not g.

---

> ### Public Comment · ~Asiri_Wijesinghe1 · 2022-02-15
> **Justification of adding 1**
>
> The justification of adding 1 is with Lemmas 1 and 2 in the appendix. Without adding 1, these lemmas would fail. For example, two distinct pairs like (H_1, W_1) and (H_2, W_2) where W_1=W_2 and H_1 $\neq$ H_2 cannot be distinguished.
>
> The function $g$ has already been defined on Page 6 in the main content of the paper. Therefore, we don't define it again in the supplementary material.

---

### Public Comment · ~Jiawei_Zhou1 · 2022-04-01
**Related Work**

Thanks for the great work and congratulations!

While reading the paper, it came to me that a previous paper of ours might be closely related in terms of the attempt to build a hierarchy (with local structures) to construct GNNs with different levels of discriminative powers beyond 1-WL, with provable justifications: [A Hierarchy of Graph Neural Networks Based on Learnable Local Features](https://arxiv.org/abs/1911.05256)

I wonder if there might be some relation in the attempt in utilizing proper neighborhood subgraphs including the edges, but it would be very interesting to have a comparison or discussion.

---

### Decision · Program_Chairs · 2022-01-20

**Decision:**

Accept (Oral)

**Comment:**

This paper proposes an efficient method for message passing that can incorporate structural information that is provably stronger than 1-WL. As compared to three strands of provably powerful (more than 1 WL) GNNs, the method has limited additional computational overhead, and can also show encouraging results on the over smoothing problem. Overall speaking, all the reviewers like this paper quite a lot, although the also raised some minor concerns. The paper also attracted some unofficial reviewers who provided quite a few related works. The authors did a good job in interacting with the reviewers and addressing their minor concerns. So, we believe the paper is worth accepting, and could be a significant work in the field of graph neural networks.